# Mastering the Game of No-Press Diplomacy via Human-Regularized Reinforcement Learning and Planning

**Anton Bakhtin**[*]  **David J Wu**[*]  **Adam Lerer**[*]  **Jonathan Gray**[*]  **Athul Paul Jacob**[*]
Meta AI          Meta AI          Meta AI          Meta AI          MIT

**Gabriele Farina**[*]          **Alexander H Miller**          **Noam Brown**
Meta AI                    Meta AI                    Meta AI

## Abstract

No-press Diplomacy is a complex strategy game involving both cooperation and competition that has served as a benchmark for multi-agent AI research. While self-play reinforcement learning has resulted in numerous successes in purely adversarial games like chess, Go, and poker, self-play alone is insufficient for achieving optimal performance in domains involving cooperation with humans. We address this shortcoming by first introducing a planning algorithm we call DiL-piKL that regularizes a reward-maximizing policy toward a human imitation-learned policy. We prove that this is a no-regret learning algorithm under a modified utility function. We then show that DiL-piKL can be extended into a self-play reinforcement learning algorithm we call RL-DiL-piKL that provides a model of human play while simultaneously training an agent that responds well to this human model. We used RL-DiL-piKL to train an agent we name Diplodocus. In a 200-game no-press Diplomacy tournament involving 62 human participants spanning skill levels from beginner to expert, two Diplodocus agents both achieved a higher average score than all other participants who played more than two games, and ranked first and third according to an Elo ratings model.

## 1 Introduction

In two-player zero-sum (2p0s) settings, principled self-play algorithms converge to a minimax equilibrium, which in a balanced game ensures that a player will not lose in expectation regardless of the opponent's strategy (Neumann, 1928). This fact has allowed self-play, even without human data, to achieve remarkable success in 2p0s games like chess (Silver et al., 2018), Go (Silver et al., 2017), poker (Bowling et al., 2015; Brown & Sandholm, 2017), and Dota 2 (Berner et al., 2019).[1] In principle, *any* finite 2p0s game can be solved via self-play given sufficient compute and memory. However, in games involving *cooperation*, self-play alone no longer guarantees good performance when playing with humans, even with *infinite* compute and memory. This is because in complex domains there may be arbitrarily many conventions and expectations for how to cooperate, of which humans may use only a small subset (Lerer & Peysakhovich, 2019). The clearest example of this is language. A self-play agent trained from scratch without human data in a cooperative game involving free-form communication channels would almost certainly not converge to using English as the medium of communication. Obviously, such an agent would perform poorly when paired with a human English speaker. Indeed, prior work has shown that naïve extensions of self-play from scratch without human data perform poorly when playing with humans or human-like agents even in dialogue-free domains that involve cooperation rather than just competition, such as the benchmark games no-press Diplomacy (Bakhtin et al., 2021) and Hanabi (Siu et al., 2021; Cui et al., 2021).

---

[*]Equal first author contribution.

[1]Dota 2 is a two-team zero-sum game, but the presence of full information sharing between teammates makes it equivalent to 2p0s. Beyond 2p0s settings, self-play algorithms have also proven successful in highly adversarial games like six-player poker (Brown & Sandholm, 2019).

Recently, (Jacob et al., 2022) introduced piKL, which models human behavior in many games better than pure behavioral cloning (BC) on human data by regularizing inference-time planning toward a BC policy. In this work, we introduce an extension of piKL, called DiL-piKL, that replaces piKL's single fixed regularization parameter $\lambda$ with a probability distribution over $\lambda$ parameters. We then show how DiL-piKL can be combined with self-play reinforcement learning, allowing us to train a strong agent that performs well with humans. We call this algorithm **RL-DiL-piKL**.

Using RL-DiL-piKL we trained an agent, Diplodocus, to play no-press Diplomacy, a difficult benchmark for multi-agent AI that has been actively studied in recent years (Paquette et al., 2019; Anthony et al., 2020; Gray et al., 2020; Bakhtin et al., 2021; Jacob et al., 2022). We conducted a 200-game no-press Diplomacy tournament with a diverse pool of human players, including expert humans, in which we tested two versions of Diplodocus using different RL-DiL-piKL settings, and other baseline agents. All games consisted of one bot and six humans, with all players being anonymous for the duration of the game. These two versions of Diplodocus achieved the top two average scores in the tournament among all 48 participants who played more than two games, and ranked first and third overall among all participants according to an Elo ratings model. Our code and models are available at `https://github.com/facebookresearch/diplomacy_cicero`.

## 2 BACKGROUND AND PRIOR WORK

Diplomacy is a benchmark 7-player mixed cooperative/competitive game featuring simultaneous moves and a heavy emphasis on negotiation and coordination. In the no-press variant of the game, there is no cheap talk communication. Instead, players only implicitly communicate through moves.

In the game, seven players compete for majority control of 34 "supply centers" (SCs) on a map. On each turn, players simultaneously choose actions consisting of an order for each of their units to hold, move, support or convoy another unit. If no player controls a majority of SCs and all remaining players agree to a draw or a turn limit is reached then the game ends in a draw. In this case, we use a common scoring system in which the score of player $i$ is $C_i^2 / \sum_{i'} C_{i'}^2$, where $C_i$ is the number of SCs player $i$ owns. A more detailed description of the rules is provided in Appendix A.

Most recent successes in no-press Diplomacy use deep learning to imitate human behavior given a corpus of human games. The first Diplomacy agent to leverage deep imitation learning was Paquette et al. (2019). Subsequent work on no-press Diplomacy have mostly relied on a similar architecture with some modeling improvements (Gray et al., 2020; Anthony et al., 2020; Bakhtin et al., 2021).

Gray et al. (2020) proposed an agent that plays an improved policy via one-ply search. It uses policy and value functions trained on human data to conduct search using regret minimization.

Several works explored applying self-play to compute improved policies. Paquette et al. (2019) applied an actor-critic approach and found that while the agent plays stronger in populations of other self-play agents, it plays worse against a population of human-imitation agents. Anthony et al. (2020) used a self-play approach based on a modification of fictitious play in order to reduce drift from human conventions. The resulting policy is stronger than pure imitation learning in both 1vs6 and 6vs1 settings but weaker than agents that use search. Most recently, Bakhtin et al. (2021) combined one-ply search based on equilibrium computation with value iteration to produce an agent called DORA. DORA achieved superhuman performance in a 2p0s version of Diplomacy without human data, but in the full 7-player game plays poorly with agents other than itself.

Jacob et al. (2022) showed that regularizing inference-time search techniques can produce agents that are not only strong but can also model human behaviour well. In no-press Diplomacy, they show that regularizing hedge (an equilibrium-finding algorithm) with a KL-divergence penalty towards a human imitation learning policy can match or exceed the human action prediction accuracy of imitation learning while being substantially stronger. KL-regularization toward human behavioral policies has previously been proposed in various forms in single- and multi-agent RL algorithms (Nair et al., 2018; Siegel et al., 2020; Nair et al., 2020), and was notably employed in AlphaStar (Vinyals et al., 2019), but this has typically been used to improve sample efficiency and aid exploration rather than to better model and coordinate with human play.

An alternative line of research has attempted to build human-compatible agents without relying on human data (Hu et al., 2020; 2021; Strouse et al., 2021). These techniques have shown some success in simplified settings but have not been shown to be competitive with humans in large-scale collaborative environments.

## 2.1 MARKOV GAMES

In this work, we focus on multiplayer Markov games (Shapley, 1953).

**Definition.** *An $n$-player Markov game $\Delta$ is a tuple $\langle S, A_1, \ldots, A_n, r_1, \ldots, r_n, p \rangle$ where $S$ is the state space, $A^i$ is the action space of player $i$ ($i = 1, \ldots, n$), $r_i : S \times A_1 \times \cdots \times A_n \to \mathbb{R}$ is the reward function for player $i$, $f : S \times A_1 \times \cdots \times A_n \to S$ is the transition function.*

The goal of each player $i$, is to choose a policy $\pi_i(s) : S \to \Delta A_i$ that maximizes the expected reward for that player, given the policies of all other players. In case of $n = 1$, a Markov game reduces to a Markov Decision Process (MDP) where an agent interacts with a fixed environment.

At each state $s$, each player $i$ simultaneously chooses an action $a_i$ from a set of actions $\mathcal{A}_i$. We denote the actions of all players other than $i$ as $\boldsymbol{a}_{-i}$. Players may also choose a probability distribution over actions, where the probability of action $a_i$ is denoted $\pi_i(s, a_i)$ or $\sigma_i(a_i)$ and the vector of probabilities is denoted $\boldsymbol{\pi}_i(s)$ or $\boldsymbol{\pi}_i$.

## 2.2 HEDGE

**Hedge** Littlestone & Warmuth (1994); Freund & Schapire (1997) is an iterative algorithm that converges to an equilibrium. We use variants of hedge for planning by using them to compute an equilibrium policy on each turn of the game and then playing that policy.

Assume that after player $i$ chooses an action $a_i$ and all other players choose actions $a_{-i}$, player $i$ receives a reward of $u_i(a_i, \boldsymbol{a}_{-i})$, where $u_i$ will come from our RL-trained value function. We denote the average reward in hindsight for action $a_i$ up to iteration $t$ as $Q^t(a_i) = \frac{1}{t} \sum_{t' \leq t} u_i(a_i, a_{-i}^{t'})$.

On each iteration $t$ of hedge, the policy $\boldsymbol{\pi}_i^t(a_i)$ is set according to $\boldsymbol{\pi}_i^t(a_i) \propto \exp\left(Q^{t-1}(a_i)/\kappa_{t-1}\right)$ where $\kappa_t$ is a temperature parameter.[2]

It is proven that if $\kappa_t$ is set to $\frac{1}{\sqrt{t}}$ then as $t \to \infty$ the *average* policy over all iterations converges to a coarse correlated equilibrium, though in practice it often comes close to a Nash equilibrium as well. In all experiments we set $\kappa_t = \frac{3\mathcal{S}_t}{10\sqrt{t}}$ on iteration $t$, where $\mathcal{S}_t$ is the observed standard deviation of the player's utility up to iteration $t$, based on a heuristic from Brown et al. (2017). A simpler choice is to set $\kappa_t = 0$, which makes the algorithm equivalent to fictitious play (Brown, 1951).

**Regret matching (RM)** (Blackwell et al., 1956; Hart & Mas-Colell, 2000) is an alternative equilibrium-finding algorithm that has similar theoretical guarantees to hedge and was used in previous work on Diplomacy Gray et al. (2020); Bakhtin et al. (2021). We do not use this algorithm but we do evaluate baseline agents that use RM.

## 2.3 DORA: SELF-PLAY LEARNING IN MARKOV GAMES

Our approach draws significantly from DORA (Bakhtin et al., 2021), which we describe in more detail here. In this approach, the authors run an algorithm that is similar to past model-based reinforcement-learning methods such as AlphaZero (Silver et al., 2018), except in place of Monte Carlo tree search, which is unsound in simultaneous-action games such as Diplomacy or other imperfect information games, it instead uses an equilibrium-finding algorithm such as hedge or RM to iteratively approximate a Nash equilibrium for the current state (i.e., one-step lookahead search). A deep neural net trained to predict the policy is used to sample plausible actions for all players to reduce the large action space in Diplomacy down to a tractable subset for the equilibrium-finding procedure, and a deep neural net trained to predict state values is used to evaluate the results of joint actions sampled by this procedure. Beginning with a policy and value network randomly initialized from scratch, a large number of self-play games are played and the resulting equilibrium policies and the improved 1-step value estimates computed on every turn from equilibrium-finding are added to a replay buffer used for subsequently improving the policy and value. Additionally, a double-oracle (McMahan et al., 2003) method was used to allow the policy to explore and discover additional actions, and the same equilibrium-finding procedure was also used at test time.

For the core update step, Bakhtin et al. (2021) propose Deep Nash Value Iteration (DNVI), a value iteration procedure similar to Nash Q-Learning (Hu & Wellman, 2003), which is a generalization

---

[2] We use $\kappa_t$ rather than $\eta$ used in Jacob et al. (2022) in order to clean up notation. $\kappa_t = 1/(\eta \cdot t)$.

of Q-learning (Watkins, 1989) from MDPs to Stochastic games. The idea of Nash-Q is to compute equilibrium policies $\sigma$ in a subgame where the actions correspond to the possible actions in a current state and the payoffs are defined using the current approximation of the value function. Bakhtin et al. (2021) propose an equivalent update that uses a state value function $V(s)$ instead of a state-action value function $Q(s, a)$:

$$\boldsymbol{V}(s) \leftarrow (1 - \alpha)\boldsymbol{V}(s) + \alpha(\boldsymbol{r} + \gamma \sum_{\boldsymbol{a'}} \sigma(\boldsymbol{a'})\boldsymbol{V}(f(s, \boldsymbol{a'}))) \tag{1}$$

where $\alpha$ is the learning rate, $\sigma(\cdot)$ is the probability of joint action in equilibrium, $\boldsymbol{a'}$ is joint action, and $f$ is the transition function. For 2p0s games and certain other game classes, this algorithm converges to a Nash equilibrium in the original stochastic game under the assumption that an exploration policy is used such that each state is visited infinitely often.

The tabular approach of Nash-Q does not scale to large games such as Diplomacy. DNVI replaces the explicit value function table and update rule in 1 with a value function parameterized by a neural network, $\boldsymbol{V}(s; \theta_v)$ and uses gradient descent to update it using the following loss:

$$\text{ValueLoss}(\theta_v) = \frac{1}{2}\left(\boldsymbol{V}(s; \theta_v) - \boldsymbol{r}(s) - \gamma \sum_{\boldsymbol{a'}} \sigma(\boldsymbol{a'})\boldsymbol{V}\left(f(s, \boldsymbol{a'}); \hat{\theta}_v\right)\right)^2 \tag{2}$$

The summation used in 2 is not feasible in games with large action spaces as the number of joint actions grow exponentially with the number of players. Bakhtin et al. (2021) address this issue by considering only a subset of actions at each step. An auxiliary function, a policy proposal network $\pi_i(s, a_i; \theta_\pi)$, models the probability that an action $a_i$ of player $i$ is in the support of the equilibrium $\sigma$. Only the top-$k$ sampled actions from this distribution are considered when solving for the equilibrium policy $\sigma$ and computing the above value loss. Once the equilibrium is computed, the equilibrium policy is also used to further train the policy proposal network using cross entropy loss:

$$\text{PolicyLoss}(\theta_\pi) = -\sum_i \sum_{a_i \in A_i} \sigma_i(a) \log \pi_i(s, a_i; \theta_\pi). \tag{3}$$

Bakhtin et al. (2021) report that the resulting agent DORA does very well when playing with other copies of itself. However, DORA performs poorly in games with 6 human human-like agents.

### 2.4 PIKL: MODELING HUMANS WITH IMITATION-ANCHORED PLANNING

*Behavioral cloning (BC)* is the standard approach for modeling human behaviors given data. Behavioral cloning learns a policy that maximizes the likelihood of the human data by gradient descent on a cross-entropy loss. However, as observed and discussed in Jacob et al. (2022), BC often falls short of accurately modeling or matching human-level performance, with BC models underperforming the human players they are trained to imitate in games such as Chess, Go, and Diplomacy. Intuitively, it might seem that initializing self-play with an imitation-learned policy would result in an agent that is both strong and human-like. Indeed, Bakhtin et al. (2021) showed improved performance against human-like agents when initializing the DORA training procedure from a human imitation policy and value, rather than starting from scratch. However, we show in subsection 5.3 that such an approach still results in policies that deviate from human-compatible equilibria.

Jacob et al. (2022) found that an effective solution was to perform search with a regularization penalty proportional to the KL divergance from a human imitation policy. This algorithm is referred to as **piKL**. The form of piKL we focus on in this paper is a variant of hedge called piKL-hedge, in which each player $i$ seeks to maximize expected reward, while at the same time playing "close" to a fixed **anchor policy** $\tau_i$. The two goals can be reconciled by defining a composite utility function that adds a penalty based on the "distance" between the player policy and their anchor policy, with coefficient $\lambda_i \in [0, \infty)$ scaling the penalty.

For each player $i$, we define $i$'s utility as a function of the agent policy $\boldsymbol{\pi}_i \in \Delta(A_i)$ given policies $\boldsymbol{\pi}_{-i}$ of all other agents:

$$\tilde{u}_{i,\lambda_i}(\boldsymbol{\pi}_i, \boldsymbol{\pi}_{-i}) := u_i(\boldsymbol{\pi}_i, \boldsymbol{\pi}_{-i}) - \lambda_i \, D_{\text{KL}}(\boldsymbol{\pi}_i \,\|\, \boldsymbol{\tau}_i) \tag{4}$$

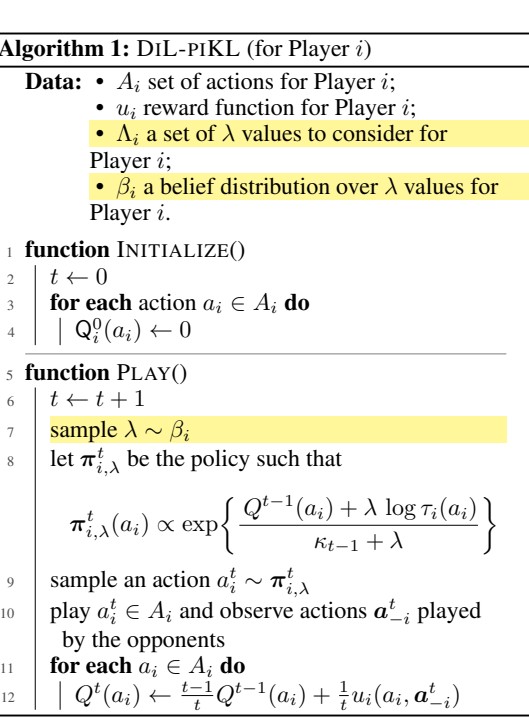

**Algorithm 1:** DIL-PIKL (for Player $i$)

**Data:** • $A_i$ set of actions for Player $i$;
• $u_i$ reward function for Player $i$;
• $\Lambda_i$ a set of $\lambda$ values to consider for Player $i$;
• $\beta_i$ a belief distribution over $\lambda$ values for Player $i$.

1 **function** INITIALIZE()
2   $t \leftarrow 0$
3   **for each** action $a_i \in A_i$ **do**
4     $Q_i^0(a_i) \leftarrow 0$

5 **function** PLAY()
6   $t \leftarrow t + 1$
7   sample $\lambda \sim \beta_i$
8   let $\boldsymbol{\pi}_{i,\lambda}^t$ be the policy such that

$$\boldsymbol{\pi}_{i,\lambda}^t(a_i) \propto \exp\left\{ \frac{Q^{t-1}(a_i) + \lambda \log \tau_i(a_i)}{\kappa_{t-1} + \lambda} \right\}$$

9   sample an action $a_i^t \sim \boldsymbol{\pi}_{i,\lambda}^t$
10   play $a_i^t \in A_i$ and observe actions $\boldsymbol{a}_{-i}^t$ played by the opponents
11   **for each** $a_i \in A_i$ **do**
12     $Q^t(a_i) \leftarrow \frac{t-1}{t} Q^{t-1}(a_i) + \frac{1}{t} u_i(a_i, \boldsymbol{a}_{-i}^t)$

Figure 1: DiL-piKL algorithm. Lines with highlights show the main differences between this algorithm and piKL-Hedge algorithm proposed in Jacob et al. (2022).

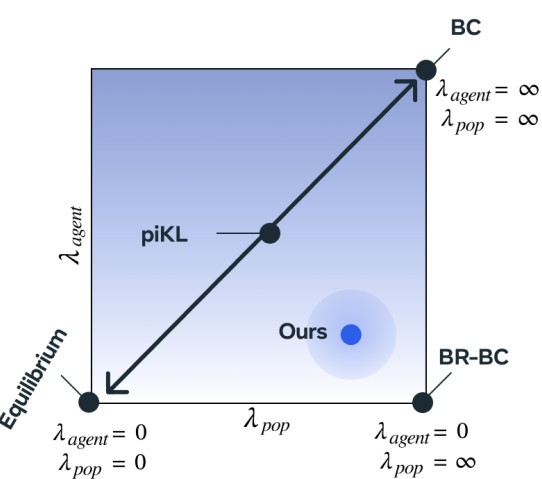

Figure 2: $\lambda_{pop}$ represents the common-knowledge belief about the $\lambda$ parameter or distribution used by all players. $\lambda_{agent}$ represents the $\lambda$ value actually used by the agent to determine its policy. By having $\lambda_{agent}$ differ from $\lambda_{pop}$, DiL-piKL interpolates between an equilibrium under the utility function $u_i$, behavioral cloning and best response to behavioral cloning policies. piKL assumed a common $\lambda$, which moved it along one axis of the space. Our agent models and coordinates with high-$\lambda$ players while playing a lower $\lambda$ itself.

This results in a modification of hedge such that on each iteration $t$, $\boldsymbol{\pi}_i^t(a_i)$ is set according to

$$\boldsymbol{\pi}_i^t(a_i) \propto \exp\left\{ \frac{Q^{t-1}(a_i) + \lambda \log \tau_i(a_i)}{\kappa_{t-1} + \lambda} \right\} \tag{5}$$

When $\lambda$ is large, the utility function is dominated by the KL-divergence term $\lambda_i D_{\mathrm{KL}}(\boldsymbol{\pi}_i \,\|\, \boldsymbol{\tau}_i)$, and so the agent will naturally tend to play a policy $\boldsymbol{\pi}_i$ close to the anchor policy $\boldsymbol{\tau}_i$. When $\lambda_i$ is small, the dominating term is the rewards $u_i(\boldsymbol{\pi}_i, \boldsymbol{a}_{-i}^t)$ and so the agent will tend to maximize reward without as closely matching the anchor policy $\boldsymbol{\tau}_i$.

## 3 DISTRIBUTIONAL LAMBDA PIKL (DIL-PIKL)

piKL trades off between the strength of the agent and the closeness to the anchor policy using a single fixed $\lambda$ parameter. In practice, we find that sampling $\lambda$ from a probability distribution each iteration produces better performance. In this section, we introduce **distributional lambda piKL (DiL-piKL)**, which replaces the single $\lambda$ parameter in piKL with a probability distribution $\beta$ over $\lambda$ values. On each iteration, each player $i$ samples a $\lambda$ value from $\beta_i$ and then chooses a policy based on Equation 5 using that sampled $\lambda$. Figure 1 highlights the difference between piKL and DiL-piKL.

One interpretation of DiL-piKL is that each choice of $\lambda$ is an agent **type**, where agent types with high $\lambda$ choose policies closer to $\boldsymbol{\tau}$ while agent types with low $\lambda$ choose policies that are more "optimal" and less constrained to a common-knowledge anchor policy. A priori, each player is randomly sampled from this population of agent types, and the distribution $\beta_i$ represents the common-knowledge uncertainty about which of the agent types player $i$ may be. Another interpretation is that piKL assumed an exponential relation between action EV and likelihood, whereas DiL-piKL results in a fatter-tailed distribution that may more robustly model different playing styles or game situations.

### 3.1 COORDINATING WITH PIKL POLICIES

While piKL and DiL-piKL are intended to model human behavior, an optimal policy in cooperative environments should be closer to a *best response* to this distribution. Selecting different $\lambda$ values for

the common-knowledge population versus the policy the agent actually plays allows us to interpolate between BC, best response to BC, and equilibrium policies (Figure 2). In practice, our agent samples from $\beta_i$ during equilibrium computation but ultimately plays a low $\lambda$ policy, modeling the fact that other players are unaware of our agent's true type.

## 3.2 Theoretical Properties of DiL-piKL

DiL-piKL can be understood as a *sampled* form of follow-the-regularized-leader (FTRL). Specifically, one can think of Algorithm 1 as an instantiation of FTRL over the Bayesian game induced by the set $\Lambda_i = \text{supp}\,\beta_i$ of types $\lambda_i$ and the regularized utilities $\tilde{u}_{i,\lambda_i}$ of each player $i$. In the appendix we show that when a player $i$ learns using DiL-piKL, the distributions $\pi_{i,\lambda}^t$ for any type $\lambda_i \in \Lambda_i$ are no-regret with respect to the regularized utilities $\tilde{u}_{i,\lambda_i}$ defined in (4). Formally:

**Theorem 1** (abridged). *Let $W$ be a bound on the maximum absolute value of any payoff in the game, and $Q_i := \frac{1}{n_i}\sum_{a\in A_i}\log\tau_i(a)$. Then, for any player $i$, type $\lambda_i \in \Lambda_i$, and number of iterations $T$, the regret cumulated can be upper bounded as*

$$\max_{\boldsymbol{\pi}\in\Delta(A_i)}\left\{\sum_{t=1}^{T}\tilde{u}_{i,\lambda_i}(\boldsymbol{\pi},\boldsymbol{a}_{-i}^t)-\tilde{u}_{i,\lambda_i}(\boldsymbol{\pi}_{i,\lambda_i}^t,\boldsymbol{a}_{-i}^t)\right\}\leq\frac{W^2}{4}\min\left\{\frac{2\log T}{\lambda_i},T\eta\right\}+\frac{\log n_i}{\eta}+\rho_{i,\lambda_i},$$

*where the game constant $\rho_{i,\lambda_i}$ is defined as $\rho_{i,\lambda_i} := \lambda_i(\log n_i + Q_i)$.*

The traditional analysis of FTRL is not applicable to DiL-piKL because the utility functions and their gradients, can be unbounded due to the nonsmoothness of the term $-\lambda_i D_{\text{KL}}(\boldsymbol{\pi}\,\|\,\boldsymbol{\tau}_i)$ that appears in the regularized utility function $\tilde{u}_{i,\lambda_i}$, and therefore a more sophisticated analysis needs to be carried out. Furthermore, even in the special case of a single type (*i.e.*, a singleton set $\Lambda_i$), where DiL-piKL coincides with piKL, the above guarantee significantly refines the analysis of piKL in two ways. First, it holds no matter the choice of stepsize $\eta > 0$, thus implying a $O(\log T/(T\lambda_i))$ regret bound without assumptions on $\eta$ other than $\eta = \Omega(1)$. Second, in the cases in which $\lambda_i$ is tiny, by choosing $\eta = \Theta(1/\sqrt{T})$ we recover a sublinear guarantee (of order $\sqrt{T}$) on the regret.

In 2p0s games, the logarithmic regret of Theorem 1 immediately implies that the *average policy* $\bar{\boldsymbol{\pi}}_{i,\lambda_i}^T := \frac{1}{T}\sum_{t=1}^{T}\boldsymbol{\pi}_{i,\lambda_i}^t$ of each player $i$ is a $\frac{C\log T}{T}$-approximate Bayes-Nash equilibrium strategy. In fact, a strong guarantee on the *last-iterate* convergence of the algorithm can be obtained too:

**Theorem 2** (abridged; Last-iterate convergence of piKL in 2p0s games). *When both players in a 2p0s game learn using DiL-piKL for $T$ iterations, their policies converge almost surely to the unique Bayes-Nash equilibrium $(\boldsymbol{\pi}_{i,\lambda_i}^*)$ of the regularized game defined by utilities $\tilde{u}_{i,\lambda_i}$ (4).*

The last-iterate guarantee stated in Theorem 2 crucially relies on the strong convexity of the regularized utilities, and conceptually belongs with related efforts in showing last-iterate convergence of online learning methods. However, a key difficulty that sets apart Theorem 2 is the fact that the learning agents observe *sampled* actions from the opponents, which makes the proof of the result (as well as the obtained convergence rate) different from prior approaches.

## 4 Description of Diplodocus

By replacing the equilibrium-finding algorithm used in DORA with DiL-piKL, we hypothesize that we can learn a strong and human-compatible policy as well as a value function that can accurately evaluate game states, assuming strong and human-like continuation policies. We call this self-play algorithm RL-DiL-piKL. We use RL-DiL-piKL to train value and policy proposal networks and use DiL-piKL during test-time search.

### 4.1 Training

Our training algorithm closely follows that of DORA, described in Section 2.3. The loss functions used are identical to DORA and the training procedure is largely the same, except in place of RM to compute the equilibrium policy $\sigma$ on each turn of a game during self-play, we use DiL-piKL with a $\lambda$ distribution and human imitation anchor policy $\tau$ that is fixed for all of training. See Appendix G for a detailed description of differences between DORA and RL-DiL-piKL.

## 4.2 TEST-TIME SEARCH

Following Bakhtin et al. (2021), at evaluation time we perform 1-ply lookahead where on each turn we sample up to 30 of the most likely actions for each player from the RL policy proposal network. However, rather than using RM to compute the equilibrium $\sigma$, we apply DiL-piKL.

As also mentioned previously in Section 3, while our agent samples $\lambda_i$ from the probability distribution $\beta_i$ when computing the DiL-piKL equilibrium, the agent chooses its own action to actually play using a fixed low $\lambda$. For all experiments, including all ablations, the agent uses the same BC anchor policy. For DiL-piKL experiments for each player $i$ we set $\beta_i$ to be uniform over $\{10^{-4}, 10^{-3}, 10^{-2}, 10^{-1}\}$ and play according to $\lambda = 10^{-4}$, except for the first turn of the game. On the first turn we instead sample from $\{10^{-2}, 10^{-1.5}, 10^{-1}, 10^{-0.5}\}$ and play according to $\lambda = 10^{-2}$, so that the agent plays more diverse openings, which more closely resemble those that humans play.

## 5 EXPERIMENTS

We first compare the performance of two variants of Diplodocus in a population of prior agents and other baseline agents. We then report results of Diplodocus playing in a tournament with humans.

## 5.1 EXPERIMENTAL SETUP

In order to measure the ability of agents to play well against a diverse set of opponents, we play many games between AI agents where each of the seven players are sampled randomly from a population of baselines (listed in Appendix C) or the agent to be tested. We report scores for each of the following algorithms against the baseline population. **Diplodocus-Low** and **Diplodocus-High** are the proposed agents that use RL-DiL-piKL during training with 2 player types $\{10^{-4}, 10^{-1}\}$ and $\{10^{-2}, 10^{-1}\}$, respectively. **DORA** is an agent that is trained via self-play and uses RM as the search algorithm during training and test-time. Both the policy and the value function are randomly initialized at the start of training. **DNVI** is similar to DORA, but the policy proposal and value networks are initialized from human BC pretraining. **DNVI-NPU** is similar to DNVI, but during training only the RL value network is updated. The policy proposal network is trained but never fed back to self-play workers, to limit drift from human conventions. The trained policy proposal is only used at test time (along with the RL value network). **BRBot** is an approximate best response to the BC policy. It was trained the same as Diplodocus, except that during training the agent plays one distinguished player each game with $\lambda = 0$ while all other players use $\lambda \approx \infty$. **SearchBot** is a one-step lookahead equilibrium search agent from (Gray et al., 2020), evaluated using their published model. **HedgeBot** is an agent similar to SearchBot (Gray et al., 2020) but using our latest architecture and using hedge rather than RM as the equilibrium-finding algorithm. **FPPI-2** and **SL** are two agents from (Anthony et al., 2020), evaluated using their published model.

After computing population scores, as a final evaluation we organized a tournament where we evaluated four agents for 50 games each in a population of online human participants. We evaluated two baseline agents, BRBot and DORA, and our new agents, Diplodocus-Low and Diplodocus-High.

In order to limit the duration of games to only a few hours, these games used a time limit of 5 minutes per turn and a stochastic game-end rule where at the beginning of each game year between 1909 and 1912 the game ends immediately with 20% chance per year, increasing in 1913 to a 40% chance. Players were not told which turn the game would end on for a specific game, but were told the distribution it was sampled from. Our agents were also trained based on this distribution.[3] Players were recruited from Diplomacy mailing lists and from `webdiplomacy.net`. In order to mitigate the risk of cheating by collusion, players were paid hourly rather than based on in-game performance. Each game had exactly one agent and six humans. The players were informed that there was an AI agent in each game, but did not know which player was the bot in each particular game. In total 62 human participants played 200 games with 44 human participants playing more than two games and 39 human participants playing at least 5 games.

---

[3]Games were run by a third-party contractor. In contradiction of the criteria we specified, the contractor ended games artificially early for the first ∼80 games played in the tournament, with end dates of 1909-1911 being more common than they should have been. We immediately corrected this problem once it was identified.

## 5.2 EXPERIMENTAL RESULTS

We first report results for our agents in the fixed population described in Appendix C. The results, shown in Table 1, show Diplodocus-Low and Diplodocus-High perform the best by a wide margin.

| Agent | Score against population |
|---|---|
| Diplodocus-Low | 29% $\pm$ 1% |
| Diplodocus-High | 28% $\pm$ 1% |
| DNVI-NPU (retrained) (Bakhtin et al., 2021) | 20% $\pm$ 1% |
| BRBot | 18% $\pm$ 1% |
| DNVI (retrained) (Bakhtin et al., 2021) | 15% $\pm$ 1% |
| HedgeBot (retrained) (Jacob et al., 2022) | 14% $\pm$ 1% |
| DORA (retrained) (Bakhtin et al., 2021) | 13% $\pm$ 1% |
| FPPI-2 (Anthony et al., 2020) | 9% $\pm$ 1% |
| SearchBot (Gray et al., 2020) | 7% $\pm$ 1% |
| SL (Anthony et al., 2020) | 6% $\pm$ 1% |

Table 1: Performance of different agents in a population of various agents. Agents above the line were retrained using our models (see Appendix E). Agents below the line were evaluated using the models and the parameters provided by the authors. The $\pm$ shows one standard error.

We next report results for the human tournament in Table 2. For each listed player, we report their average score, Elo rating, and rank within the tournament based on Elo among players who played at least 5 games. Elo ratings were computed using a standard generalization of BayesElo (Coulom, 2005) to multiple players (Hunter, 2004) (see Appendix H for details). This gives similar rankings as average score, but also attempts to correct for both the average strength of the opponents, since some games may have stronger or weaker opposition, as well as for which of the seven European powers a player was assigned in each game, since some starting positions in Diplomacy are advantaged over others. To regularize the model, a weak Bayesian prior was applied such that each player's rating was normally distributed around 0 with a standard deviation of around 350 Elo.

| | Rank | Elo | Avg Score | # Games |
|---|---|---|---|---|
| **Diplodocus-High** | 1 | 181 | 27% $\pm$ 4% | 50 |
| Human | 2 | 162 | 25% $\pm$ 6% | 13 |
| **Diplodocus-Low** | 3 | 152 | 26% $\pm$ 4% | 50 |
| Human | 4 | 138 | 22% $\pm$ 9% | 7 |
| Human | 5 | 136 | 22% $\pm$ 3% | 57 |
| **BRBot** | 6 | 119 | 23% $\pm$ 4% | 50 |
| Human | 7 | 102 | 18% $\pm$ 8% | 8 |
| Human | 8 | 96 | 17% $\pm$ 3% | 51 |
| . . . | . . . | . . . | . . . | . . . |
| **DORA** | 32 | -20 | 13% $\pm$ 3% | 50 |
| . . . | . . . | . . . | . . . | . . . |
| Human | 43 | -187 | 1% $\pm$ 1% | 7 |

Table 2: Performance of four different agents in a population of human players, ranked by Elo, among all 43 participants who played at least 5 games. The $\pm$ shows one standard error.

The results show that Diplodocus-High performed best among all the humans by both Elo and average score. Diplodocus-Low followed closely behind, ranking second according to average score and third by Elo. BRBot performed relatively well, but ended ranked below that of both DiL-piKL agents and several humans. DORA performed relatively poorly.

Two participants achieved a higher average score than the Diplodocus agents, a player averaging 35% but who only played two games, and a player with a score of 29% who played only one game.

We note that given the large statistical error margins, the results in Table 2 do not conclusively demonstrate that Diplodocus outperforms the best human players, nor do they alone demonstrate an unambiguous separation between Diplodocus and BRBot. However, the results do indicate that Diplodocus performs at least at the level of expert players in this population of players with diverse

skill levels. Additionally, the superior performance of both Diplodocus agents compared to BRBot is consistent with the results from the agent population experiments in Table 1.

In addition to the tournament, we asked three expert human players to evaluate the strength of the agents in the tournament games based on the quality of their actions. Games were presented to these experts with anonymized labels so that the experts were *not* aware of which agent was which in each game when judging that agent's strategy. All the experts picked a Diplodocus agent as the strongest agent, though they disagreed about whether Diplodocus-High or Diplodocus-Low was best. Additionally, all experts indicated one of the Diplodocus agents as the one they would most like to cooperate with in a game. We provide detailed responses in Appendix B.

### 5.3 RL TRAINING COMPARISON

Figure 3 compares different RL agents across the course of training. To simplify the comparison, we vary the training methods for the value and policy proposal networks, but use the same search setting at evaluation time.

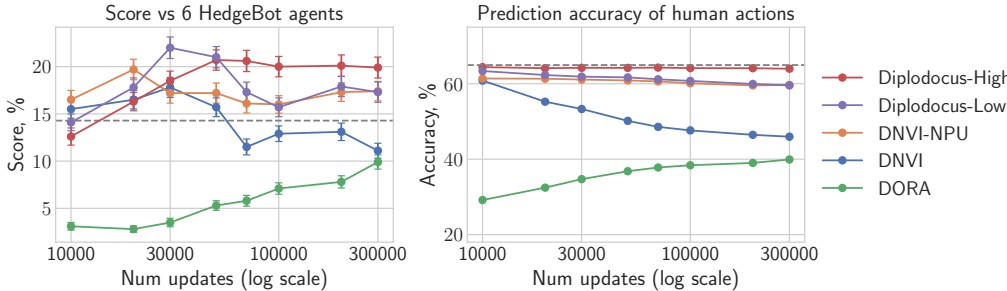

Figure 3: Performance of different agents as a function of the number of RL training steps. **Left:** Scores against 6 human-like HedgeBot agents. The gray dotted line at score $1/7 \approx 14.3\%$ corresponds to tying HedgeBot. The error bars show one standard error. **Right:** Order prediction accuracy of each agent's raw RL policy on a held-out set of human games. The gray dotted line corresponds to the behavioral cloning policy. **Overall:** Diplodocus-High achieves a high score while also maintaining high prediction accuracy. Unregularized agents DNVI and DORA do far worse on both metrics.

As a proxy for agent strength, we measure the average score of an agent vs 6 copies of HedgeBot. As a proxy for modeling humans, we compute prediction accuracy of human moves on a validation dataset of roughly 630 games held out from training of the human BC model, i.e., how often the most probable action under the policy corresponds to the one chosen by a human. Similar to Bakhtin et al. (2021), we found that agents without biasing techniques (DORA and DNVI) diverge from human play as training progress. By contrast, Diplodocus-High achieves significant improvement in score while keeping the human prediction accuracy high.

## 6 DISCUSSION

In this work we describe RL-DiL-piKL and use it to train an agent for no-press Diplomacy that placed first in a human tournament. We ascribe Diplodocus's success in Diplomacy to two ideas.

First, DiL-piKL models a population of player types with different amounts of regularization to a human policy while ultimately playing a strong (low-$\lambda$) policy itself. This improves upon simply playing a best response to a BC policy by accounting for the fact that humans are less likely to play highly suboptimal actions and by reducing overfitting of the best response to the BC policy. Second, incorporating DiL-piKL in self-play allows us to learn an accurate value function in a diversity of situations that arise from strong and human-like players. Furthermore, this value assumes a human continuation policy that makes fewer blunders than the BC policy, allowing us to correctly estimate the values of positions that require accurate play (such as stalemate lines).

In conclusion, combining human imitation, planning, and RL presents a promising avenue for building agents for complex cooperative and mixed-motive environments. Further work could explore regularized search policies that condition on more complex human behavior, including dialogue.

## ACKNOWLEDGMENTS

We thank the anonymous reviewers for the stimulating discussion and for helping improve the paper.

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

## A    DESCRIPTION OF DIPLOMACY

The rules of no-press Diplomacy are complex; a full description is provided by Paquette et al. (2019). No-press Diplomacy is a seven-player zero-sum board game in which a map of Europe is divided into 75 provinces. 34 of these provinces contain supply centers (SCs), and the goal of the game is for a player to control a majority (18) of the SCs. Each players begins the game controlling three or four SCs and an equal number of units.

The game consists of three types of phases: movement phases in which each player assigns an order to each unit they control, retreat phases in which defeated units retreat to a neighboring province, and adjustment phases in which new units are built or existing units are destroyed.

During a movement phase, a player assigns an order to each unit they control. A unit's order may be to hold (defend its province), move to a neighboring province, convoy a unit over water, or support a neighboring unit's hold or move order. Support may be provided to units of any player. We refer to a tuple of orders, one order for each of a player's units, as an **action**. That is, each player chooses one action each turn. There are an average of 26 valid orders for each unit (Paquette et al., 2019), so the game's branching factor is massive and on some turns enumerating all actions is intractable.

Importantly, all actions occur simultaneously. In live games, players write down their orders and then reveal them at the same time. This makes Diplomacy an imperfect-information game in which an optimal policy may need to be stochastic in order to prevent predictability.

Diplomacy is designed in such a way that cooperation with other players is almost essential in order to achieve victory, even though only one player can ultimately win.

A game may end in a draw on any turn if all remaining players agree. Draws are a common outcome among experienced players because players will often coordinate to prevent any individual from reaching 18 centers. The two most common scoring systems for draws are **draw-size scoring (DSS)**, in which all surviving players equally split a win, and **sum-of-squares scoring (SoS)**, in which player $i$ receives a score of $\frac{C_i^2}{\sum_{j \in \mathcal{N}} C_j^2}$, where $C_i$ is the number of SCs that player $i$ controls (Fogel, 2020). Throughout this paper we use SoS scoring except in anonymous games against humans where the human host chooses a scoring system.

## B    EXPERT EVALUATION OF THE AGENTS

The anonymous format of the tournament aimed at reducing possible biases of players towards the agent, e.g., trying to collectively eliminate the agents as targeting the agent is a simple way to break the symmetry. At the same time a significant property of Diplomacy is knowing the play styles of different players and using this knowledge to make decision of whom to trust and whom to chose as an ally. To evaluates this aspect of the game play we asked for qualitative feedback from three Diplomacy experts. Each player was given 7 games (one per power) from each of the 4 different agents that played in the tournament. The games evaluated by each expert were disjoint from the games evaluated by the other experts. The games were anonymized such that the experts were not able to tell which agent played in the game based on the username or from the date. We asked a few

questions about the game play of each agent independently and then asked the experts to choose the best agent for strength and human-like behavior. The experts referred to the agents as Agent1, ..., Agent4, but we de-anonymized the agents in the answers below.

## B.1 OVERALL

WHAT IS THE STRONGEST AGENT?

**Expert 1**   I think Diplodocus-Low was the strongest, then BRBot closely followed by Diplodocus-High. DORA is a distant third.

**Expert 2**   Diplodocus-High.

**Expert 3**   Diplodocus-Low. This feels stronger than a human in certain ways while still being very human-like.

WHAT IS THE MOST HUMAN-LIKE/BOT-LIKE AGENT?

**Expert 1**   Most human-like is Diplodocus-High. A boring human, but a human nonetheless. Diplodocus-Low is not far behind, then BRBot and DORA both of which are very non-human albeit in very different ways.

**Expert 2**   Diplodocus-High.

**Expert 3**   Diplodocus-Low.

WHAT IS THE AGENT YOU'D LIKE TO COOPERATE WITH?

**Expert 1**   This is the most interesting question. I think Diplodocus-Low, because I like how it plays - we'd "vibe" - but also because I think it is quite predictable in what motivates it to change alliances. That's potentially exploitable, even with the strong tactics it has. I'd least like to work with Diplodocus-High as it seems to be very much in it for itself. I suspect it would be quite unpleasant to play against as it is tactically excellent and seems hard to work with.

I'd love to be on a board with DORA, as I'd expect my chances to solo to go up dramatically! It would be a very fun game so long as you weren't on the receiving end of some of its play.

**Expert 2**   Diplodocus-High.

**Expert 3**   Diplodocus-Low. Diplodocus-High is also strong, but seems much less interesting to play with, because of the way it commits to alliances without taking into account who is actually willing to work with it. This limits what a human can do to change their situation quite a lot and would be fairly frustrating in the position of a neighbour being attacked by it.

BRBot and DORA feel too weak to be particularly interesting.

## B.2 DORA

HOW WOULD YOU EVALUATE THE OVERALL STRENGTH OF THE AGENT?

**Expert 1**   Not great. There's a lot to criticize here - from bad opening play (Russia = bonkers), to poor defense (Turkey) and just generally bad tactics and strategy compared to the other agents (France attacking Italy when Italy is their only ally was an egregious example of this).

**Expert 2**   Very weak. Seemed to invite its own demise with the way it routinely picked fights in theaters it had no business being in and failing to cooperate with neighbors

**Expert 3**   Poor. It is bad at working with players, and it makes easily avoidable blunders even when working alone.

HOW WOULD YOU EVALUATE THE ABILITY OF THE AGENT TO COOPERATE?

**Expert 1**   It seems to make efforts, but it also seems to misjudge what humans are likely to do. There's indicative support orders and they're pretty good, but it also doesn't seem to understand or account for vindictiveness over always playing best. The Turkey game where it repeatedly seems to expect Russia to not attack is an example of this.

**Expert 2**   Poor. Seemed to pick fights without seeing or soliciting support necessary to win, failed to support potential allies in useful ways to take advantage of their position.

**Expert 3**   Middling to Poor. It very occasionally enters good supports but it often enters bad ones, and has a habit of attacking too many people at once (and not considering that attacking those people will turn them against it). It has a habit of annoying many players and doing badly as a result.

### B.3   BRBOT

HOW WOULD YOU EVALUATE THE OVERALL STRENGTH OF THE AGENT?

**Expert 1**   The agent has solid, at least human level tactics and clearly sees opportunities to advance and acts accordingly. Sometimes this is to the detriment of the strategic position, but the balance is fair given the gunboat nature of the games. Overall, the bot feels naturally to be in the "better than average human" range rather than super-human, but the results indicate that it performs at a higher level than the "feeling" it gives. It has a major opportunity for improvement, discussed in the next point.

**Expert 2**   Overall, seemed fairly weak and seemed to be able to succeed most frequently when benefiting from severe mistakes from neighboring agents. That being said it was able to exploit those mistakes somewhat decently in some cases and at least grow to some degree off of it.

**Expert 3**   Middling. It is tactically strong when not having to work with other players and when it has a considerable number of units, but is quite weak when attempting to cooperate with other players. Its defensive strength varies quite significantly too, possibly also based on unit count - when it had relatively few units it missed very obvious defensive tactics.

HOW WOULD YOU EVALUATE THE ABILITY OF THE AGENT TO COOPERATE?

**Expert 1**   The bot is hyperactively trying to coordinate and signal to the other players that it wants to work with them. Sometimes this is in the form of ridiculous orders that probably indicate desperation more than a mutually beneficial alliance, and this backfires as you may expect. At its best it makes exceptional signaling moves (RUSSIA game[4]: War - Mos in Fall 1901 is exceptional) but at worst it is embarrassingly bad and leads to it getting attacked (TURKEY game[5]: supporting convoys from Gre - Smy or supporting other powers moving to Armenia). The other weakness is that it tends to make moves like these facing all other powers - this is not optimal as indicating to all other powers that you want to work with them is equivalent to not indicating anything at all - if anything it seems a little duplicitous. This is especially true when the bot is still inclined to stab when the opportunity presents itself, which means the signaling is superficial and unlikely to work repeatedly. Overall, the orders show the ability to cooperate, signal, and work together, but the hyperactivity of the bot is limiting the effectiveness of the tools to achieve the best results.

**Expert 2**   Poor. Random support orders seemed to be thrown without an overarching strategy behind them. Moves didn't seem to suggest long term thoughts of collaboration.

**Expert 3**   Poor. When attempting to work with another player, it almost always gives them the upper hand, and even issues supports that suggest it is okay with that player taking its SCs when it should not be. It sometimes matches supports to human moves, but does not seem to do this very often. The nonsensical supports are much more common.

---

[4]https://webdiplomacy.net/board.php?gameID=400742
[5]https://webdiplomacy.net/board.php?gameID=400741

## B.4 DIPLODOCUS-HIGH

HOW WOULD YOU EVALUATE THE OVERALL STRENGTH OF THE AGENT?

**Expert 1** The tactics are unadventurous and sometimes seem below human standards (for example, the train of army units in the Italy game; the whole Turkey game) but conversely they also have a longer view of the game (see also: Italy game - the trained bounces don't matter strategically). There's less nonsense too; if I were to sum the bot up in two words it would be "practical" and "boring".

**Expert 2** Seemed to be strong. Wrote generally good tactical orders, showed generally good strategic sense. Showed discipline and a willingness to let allies survive in weak positions while having units that could theoretically stab for dots with ease remaining right next to that weak ally.

There were some highly questionable moments as both Italy and France early on in 1901 strategy which seemed to heavily harm their ability to get out of the box.

The Austrian game was particularly impressive in terms of its ability to handle odd scenarios and achieve the solo despite receiving pressure on multiple occasions on multiple fronts.

**Expert 3** Generally strong. It is good at signalling and forming alliances, is tactically strong when in its favoured alliance, and is especially strong when ahead. Its main weakness seems to be an inability to adapt - if its favoured alliance is declined, it will often keep trying to 'pitch' that same alliance instead of working towards alternatives.

HOW WOULD YOU EVALUATE THE ABILITY OF THE AGENT TO COOPERATE?

**Expert 1** Low. It doesn't put much effort into this. The French game, for example, the bot just seems to accept it is being attacked and fight through it. It's so boring and tactical and shows little care for cooperation. Many great gunboat players do this but it will not hold up in press games. What it does seem to do is capitalize on other player's mistakes - see the Austrian game where it sneaks into Scandinavia and optimizes to get to 18 (there can't be a lot of training data for that!).

**Expert 2** Very strong ability to cooperate as seen in the Turkish game, but in other games seemed to try and pick fights against the entire world in ways that were ultimately self-defeating.

**Expert 3** Good. It can work well with human players, matching supports and even using signalling supports in ways humans would. It frequently attempts to side with a player who is attacking it, though, so it seems to have a problem with identifying which player to work with.

## B.5 DIPLODOCUS-LOW

HOW WOULD YOU EVALUATE THE OVERALL STRENGTH OF THE AGENT?

**Expert 1** Exceptional. Very strong tactics and a clear directionality to what it does - it seems to understand what the strategic value of a position is and it acts with efficiency to achieve the strategic goals. It has great results (time drawn out of a few wins!) but also fights back from "losing" positions extremely well which makes it quantifiably strong, but it also just plays a beautiful and effective game. Very strong indeed. It does sometimes act too aggressively for tournament play (Austria is the example where this came home to roost) - the high risk/reward plays are generally but not always correct in single games, but for tournament play it goes for broke a bit too much (This is outside the scope of the agent I suspect, as it is playing to scoring system not tournament scoring system). Against human players who may not see the longer term impact of their play, it results in games like this one... which is ugly both for Austria and for everyone else except Turkey.

**Expert 2** Very weak. Seemed to abandon its own position in many cases to pursue questionable adventures. Sometimes they worked out but generally they failed, resulting in things like a Germany under siege holding Edi while they as England are off in Portugal and are holding onto their home centers only because FG were under siege by the south.

**Expert 3**  Very strong. It can signal alliances very well and generally chooses the correct allies, seems strong tactically even on defence, and makes some plays you would not expect from a human player but which are outright stronger than a human player would make.

HOW WOULD YOU EVALUATE THE ABILITY OF THE AGENT TO COOPERATE?

**Expert 1**  Pretty good. It sends signaling moves and makes efforts to support other players quite a lot (see in particular Russia). I particularly like the skills being shown to work together tactically and try and support other units - this is both effective and quite human. This is my favorite bot by some distance when it comes to cooperating with the other players. There is a weakness in that it does seem to reassess alliances every turn, which means sometimes the excellent work indicating and supporting is undone without getting the chance to realize the gains (Examples with Russia and Italy).

**Expert 2**  Poor. Didn't seem to give meaningful support orders when they would have helped and gave plenty of meaningless signaling supports and some questionable ones like supporting the English into SKA in F1901 as Germany among other oddities

**Expert 3**  Good. It signals alliances in very human ways, through clear signalling builds, accurate support moves where it makes sense, and support holds otherwise. It also seems to match supports with its allies well.

## C  POPULATION BASED EVALUATION

In general-sum games like Diplomacy, winrate in head-to-head matches against a previous version of an agent may not be as informative because of nontransitivity between agents. For example, exploitative agents such as best-response-to-BC may do particularly well against BC or other pure imitation-learning agents, and less well against all other agents. Additionally, Bakhtin et al. (2021) found that a pair of independently and equally-well-trained RL agents may each appear very weak in a population composed of the other due to converging to incompatible equilibria. Many agents also varied significantly in how well they performed against other search-based agents.

Therefore, we resort to playing against a population of previously training agents as was done in Jacob et al. (2022), intended to measure more broadly how well an agent does on average against a wider suite of various human-like agents.

More precisely, we define a fixed set of baseline agents as a population. To determine an agent's average population score, we add that agent into the population and then play games where in each game, all 7 players are uniformly randomly sampled from the population with replacement, keeping only games where the agent to be tested was sampled at least once. Note that unlike Jacob et al. (2022), we run a separate population test for each new agent to be tested, rather than combining all agents to be tested within a single population.

For the experiments in Table 1 and subsection 5.1 we used the following 8 baseline agents:

- An single-turn BR agent that assumes everyone else plays the BC policy.
- An agent doing RM search with BC policy and value functions. We use 2 copies of this agent trained on different subsets of data.
- DiL-piKL agent with BC policy and value functions. We use 4 different versions of this data with different training data and model architecture.
- DiL-piKL agent where the policy and value functions are trained with self-play with Reinforced-PiKL with high lambda ($\lambda = 3 \times 10^{-2}$).

For the experiments in this paper we used 1000 games for each such population test.

## D  THEORETICAL PROPERTIES OF DiL-piKL

In this section we study the last-iterate convergence of DiL-piKL, establishing that in two-player zero-sum games DiL-piKL converges to the (unique) Bayes-Nash equilibrium of the regularized

Bayesian game. As a corollary (in the case in which each player has exactly one type), we conclude that piKL converges to the Nash equilibrium of the regularized game in two-player zero-sum games. We start from a technical result. In all that follows, we will always let $\boldsymbol{u}_i^t$ be a shorthand for the vector $(u_i(a, \boldsymbol{a}_{-i}^t))_{a \in A_i}$.

**Lemma 1.** *Fix any player $i$, $\lambda_i \in \Lambda_i$, and $t \geq 1$. For all $\boldsymbol{\pi}, \boldsymbol{\pi}' \in \Delta(A_i)$, the iterates $\boldsymbol{\pi}_{i,\lambda_i}^t$ and $\boldsymbol{\pi}_{i,\lambda_i}^{t+1}$ defined in Line 8 of Algorithm 1 satisfy*

$$\left\langle \frac{\eta}{\eta \lambda_i t + 1} \left(-\boldsymbol{u}_i^t + \lambda_i \nabla \phi(\boldsymbol{\pi}_{i,\lambda_i}^t) - \lambda_i \nabla \phi(\boldsymbol{\tau}_i)\right) + \nabla \phi(\boldsymbol{\pi}_{i,\lambda_i}^{t+1}) - \nabla \phi(\boldsymbol{\pi}_{i,\lambda_i}^t), \boldsymbol{\pi} - \boldsymbol{\pi}' \right\rangle = 0.$$

*Proof.* If $t = 1$, then the results follows from direct inspection: $\boldsymbol{\pi}_{i,\lambda_i}^1$ is the uniform policy (and so $\langle \nabla \phi(\boldsymbol{\pi}_{i,\lambda_i}^1), \boldsymbol{\pi} - \boldsymbol{\pi}' \rangle = 0$ for any $\boldsymbol{\pi}, \boldsymbol{\pi}' \in \Delta(A_i)$, and so the statement reduces to the first-order optimality conditions for the problem $\boldsymbol{\pi}_{i,\lambda_i}^2 = \arg \max_{\boldsymbol{\pi} \in \Delta(A_i)} \{-\phi(\boldsymbol{\pi})/\eta + \langle \boldsymbol{u}_i^1, \boldsymbol{\pi} \rangle - \lambda_i D_{\mathrm{KL}}(\boldsymbol{\pi} \| \boldsymbol{\tau}_i)\}$. So, we now focus on the case $t \geq 2$. The iterates $\boldsymbol{\pi}_{i,\lambda_i}^{t+1}$ and $\boldsymbol{\pi}_{i,\lambda_i}^t$ produced by DiL-piKL are respectively the solutions to the optimization problem

$$\boldsymbol{\pi}_{i,\lambda_i}^{t+1} = \arg \max_{\boldsymbol{\pi} \in \Delta(A_i)} \left\{-\frac{\phi(\boldsymbol{\pi})}{\eta t} + \langle \bar{\boldsymbol{U}}_i^t, \boldsymbol{\pi} \rangle - \lambda_i \, D_{\mathrm{KL}}(\boldsymbol{\pi} \| \boldsymbol{\tau}_i)\right\},$$

$$\boldsymbol{\pi}_{i,\lambda_i}^t = \arg \max_{\boldsymbol{\pi} \in \Delta(A_i)} \left\{-\frac{\phi(\boldsymbol{\pi})}{\eta(t-1)} + \langle \bar{\boldsymbol{U}}_i^{t-1}, \boldsymbol{\pi} \rangle - \lambda_i \, D_{\mathrm{KL}}(\boldsymbol{\pi} \| \boldsymbol{\tau}_i)\right\},$$

where we let the averages utility vectors be

$$\bar{\boldsymbol{U}}_i^{t-1} := \frac{1}{t-1} \sum_{t'=1}^{t-1} \boldsymbol{u}_i^{t'}, \qquad \bar{\boldsymbol{U}}_i^t := \frac{1}{t} \sum_{t'=1}^{t} \boldsymbol{u}_i^{t'}.$$

Since the regularizing function negative entropy $\phi$ is Legendre, the policies $\boldsymbol{\pi}_{i,\lambda_i}^{t+1}$ and $\boldsymbol{\pi}_{i,\lambda_i}^t$ are in the relative interior of the probability simplex, and therefore the first-order optimality conditions for $\boldsymbol{\pi}_{i,\lambda_i}^{t+1}$ and $\boldsymbol{\pi}_{i,\lambda_i}^t$ are respectively

$$\left\langle -\bar{\boldsymbol{U}}_i^t + \lambda_i \nabla \phi(\boldsymbol{\pi}_{i,\lambda_i}^{t+1}) - \lambda_i \nabla \phi(\boldsymbol{\tau}_i) + \frac{1}{\eta t} \nabla \phi(\boldsymbol{\pi}_{i,\lambda_i}^{t+1}), \boldsymbol{\pi} - \boldsymbol{\pi}' \right\rangle = 0 \quad \forall \boldsymbol{\pi}, \boldsymbol{\pi}' \in \Delta(A_i),$$
(6)

$$\left\langle -\bar{\boldsymbol{U}}_i^{t-1} + \lambda_i \nabla \phi(\boldsymbol{\pi}_{i,\lambda_i}^t) - \lambda_i \nabla \phi(\boldsymbol{\tau}_i) + \frac{1}{\eta(t-1)} \nabla \phi(\boldsymbol{\pi}_{i,\lambda_i}^t), \boldsymbol{\pi} - \boldsymbol{\pi}' \right\rangle = 0 \quad \forall \boldsymbol{\pi}, \boldsymbol{\pi}' \in \Delta(A_i).$$

Taking the difference between the equalities, we find

$$\left\langle -\bar{\boldsymbol{U}}_i^t + \bar{\boldsymbol{U}}_i^{t-1} + \left(\lambda_i + \frac{1}{\eta t}\right) \nabla \phi(\boldsymbol{\pi}_{i,\lambda_i}^{t+1}) - \left(\lambda_i + \frac{1}{\eta(t-1)}\right) \nabla \phi(\boldsymbol{\pi}_{i,\lambda_i}^t), \boldsymbol{\pi} - \boldsymbol{\pi}' \right\rangle = 0$$

We now use the fact that

$$\bar{\boldsymbol{U}}_i^t - \bar{\boldsymbol{U}}_i^{t-1} = -\frac{1}{t-1} \bar{\boldsymbol{U}}_i^t + \frac{1}{t-1} \boldsymbol{u}_i^t.$$

to further write

$$\left\langle \frac{1}{t-1}(-\boldsymbol{u}_i^t + \bar{\boldsymbol{U}}_i^t) + \left(\lambda_i + \frac{1}{\eta t}\right) \nabla \phi(\boldsymbol{\pi}_{i,\lambda_i}^{t+1}) - \left(\lambda_i + \frac{1}{\eta(t-1)}\right) \nabla \phi(\boldsymbol{\pi}_{i,\lambda_i}^t), \boldsymbol{\pi} - \boldsymbol{\pi}' \right\rangle = 0 \quad (7)$$

From Equation (6) we find

$$\langle \bar{\boldsymbol{U}}_i^t, \boldsymbol{\pi} - \boldsymbol{\pi}' \rangle = \left\langle \lambda_i \nabla \phi(\boldsymbol{\pi}_{i,\lambda_i}^{t+1}) - \lambda_i \nabla \phi(\boldsymbol{\tau}_i) + \frac{1}{\eta t} \nabla \phi(\boldsymbol{\pi}_{i,\lambda_i}^{t+1}), \boldsymbol{\pi} - \boldsymbol{\pi}' \right\rangle$$

and so, plugging back the previous relationship in Equation (7) we can write, for all $\boldsymbol{\pi}, \boldsymbol{\pi}' \in \Delta(A_i)$,

$$
\begin{aligned}
0 = \Big\langle &\frac{1}{t-1}\Big(-\boldsymbol{u}_i^t + \lambda_i \nabla\phi(\boldsymbol{\pi}_{i,\lambda_i}^{t+1}) - \lambda_i \nabla\phi(\boldsymbol{\tau}_i) + \frac{1}{\eta t}\nabla\phi(\boldsymbol{\pi}_{i,\lambda_i}^{t+1})\Big) + \Big(\lambda_i + \frac{1}{\eta t}\Big)\nabla\phi(\boldsymbol{\pi}_{i,\lambda_i}^{t+1}) \\
&\hspace{5cm} -\Big(\lambda_i + \frac{1}{\eta(t-1)}\Big)\nabla\phi(\boldsymbol{\pi}_{i,\lambda_i}^t), \boldsymbol{\pi} - \boldsymbol{\pi}'\Big\rangle \\
= \Big\langle &\frac{1}{t-1}\Big(-\boldsymbol{u}_i^t + \lambda_i \nabla\phi(\boldsymbol{\pi}_{i,\lambda_i}^{t+1}) - \lambda_i \nabla\phi(\boldsymbol{\tau}_i)\Big) + \Big(\lambda_i + \frac{1}{\eta(t-1)}\Big)\nabla\phi(\boldsymbol{\pi}_{i,\lambda_i}^{t+1}) \\
&\hspace{5cm} -\Big(\lambda_i + \frac{1}{\eta(t-1)}\Big)\nabla\phi(\boldsymbol{\pi}_{i,\lambda_i}^t), \boldsymbol{\pi} - \boldsymbol{\pi}'\Big\rangle \\
= \Big\langle &\frac{1}{t-1}\Big(-\boldsymbol{u}_i^t + \lambda_i \nabla\phi(\boldsymbol{\pi}_{i,\lambda_i}^t) - \lambda_i \nabla\phi(\boldsymbol{\tau}_i)\Big) + \frac{\eta\lambda_i t + 1}{\eta(t-1)}\nabla\phi(\boldsymbol{\pi}_{i,\lambda_i}^{t+1}) \\
&\hspace{5cm} -\frac{\eta\lambda_i t + 1}{\eta(t-1)}\nabla\phi(\boldsymbol{\pi}_{i,\lambda_i}^t), \boldsymbol{\pi} - \boldsymbol{\pi}'\Big\rangle.
\end{aligned}
$$

Dividing by $(\eta\lambda_i t + 1)/(\eta(t-1))$ yields the statement. $\qquad\square$

**Corollary 1.** *Fix any player $i$, $\lambda_i \in \Lambda_i$, and $t \geq 1$. For all $\boldsymbol{\pi} \in \Delta(A_i)$, the iterates $\boldsymbol{\pi}_{i,\lambda_i}^t$ and $\boldsymbol{\pi}_{i,\lambda_i}^{t+1}$ defined in Line 8 of Algorithm 1 satisfy*

$$
\begin{aligned}
\Big\langle -\boldsymbol{u}_i^t &+ \lambda_i \nabla\phi(\boldsymbol{\pi}_{i,\lambda_i}^t) - \lambda_i \nabla\phi(\boldsymbol{\tau}_i), \boldsymbol{\pi} - \boldsymbol{\pi}_{i,\lambda_i}^{t+1}\Big\rangle \\
&= \Big(\lambda_i t + \frac{1}{\eta}\Big)\Big(D_{\mathrm{KL}}(\boldsymbol{\pi} \,\|\, \boldsymbol{\pi}_{i,\lambda_i}^{t+1}) - D_{\mathrm{KL}}(\boldsymbol{\pi} \,\|\, \boldsymbol{\pi}_{i,\lambda_i}^t) + D_{\mathrm{KL}}(\boldsymbol{\pi}_{i,\lambda_i}^{t+1} \,\|\, \boldsymbol{\pi}_{i,\lambda_i}^t)\Big).
\end{aligned}
$$

*Proof.* Since Lemma 1 holds for all $\boldsymbol{\pi}, \boldsymbol{\pi}' \in \Delta(A_i)$, we can in particular set $\boldsymbol{\pi}' = \boldsymbol{\pi}_{i,\lambda_i}^{t+1}$, and obtain

$$
\begin{aligned}
\frac{\eta}{\eta\lambda_i t + 1}\Big\langle -\boldsymbol{u}_i^t &+ \lambda_i \nabla\phi(\boldsymbol{\pi}_{i,\lambda_i}^t) - \lambda_i \nabla\phi(\boldsymbol{\tau}_i), \boldsymbol{\pi} - \boldsymbol{\pi}_{i,\lambda_i}^{t+1}\Big\rangle \\
&\hspace{2cm} + \Big\langle \nabla\phi(\boldsymbol{\pi}_{i,\lambda_i}^{t+1}) - \nabla\phi(\boldsymbol{\pi}_{i,\lambda_i}^t), \boldsymbol{\pi} - \boldsymbol{\pi}_{i,\lambda_i}^{t+1}\Big\rangle = 0. \quad (8)
\end{aligned}
$$

Using the three-point identity

$$
\Big\langle \nabla\phi(\boldsymbol{\pi}_{i,\lambda_i}^{t+1}) - \nabla\phi(\boldsymbol{\pi}_{i,\lambda_i}^t), \boldsymbol{\pi} - \boldsymbol{\pi}_{i,\lambda_i}^{t+1}\Big\rangle = D_{\mathrm{KL}}(\boldsymbol{\pi} \,\|\, \boldsymbol{\pi}_{i,\lambda_i}^t) - D_{\mathrm{KL}}(\boldsymbol{\pi} \,\|\, \boldsymbol{\pi}_{i,\lambda_i}^{t+1}) - D_{\mathrm{KL}}(\boldsymbol{\pi}_{i,\lambda_i}^{t+1} \,\|\, \boldsymbol{\pi}_{i,\lambda_i}^t)
$$

in Equation (8) yields

$$
\begin{aligned}
D_{\mathrm{KL}}(\boldsymbol{\pi} \,\|\, \boldsymbol{\pi}_{i,\lambda_i}^{t+1}) = D_{\mathrm{KL}}(\boldsymbol{\pi} \,\|\, \boldsymbol{\pi}_{i,\lambda_i}^t) &- D_{\mathrm{KL}}(\boldsymbol{\pi}_{i,\lambda_i}^{t+1} \,\|\, \boldsymbol{\pi}_{i,\lambda_i}^t) \\
&+ \frac{\eta}{\eta\lambda_i t + 1}\Big\langle -\boldsymbol{u}_i^t + \lambda_i \nabla\phi(\boldsymbol{\pi}_{i,\lambda_i}^t) - \lambda_i \nabla\phi(\boldsymbol{\tau}_i), \boldsymbol{\pi} - \boldsymbol{\pi}_{i,\lambda_i}^{t+1}\Big\rangle.
\end{aligned}
$$

Multiplying by $\lambda_i t + 1/\eta$ yields the statement. $\qquad\square$

## D.1 REGRET ANALYSIS

Let $\tilde{u}_{i,\lambda_i}^t$ be the regularized utility of agent type $\lambda_i \in \Lambda_i$

$$
\tilde{u}_{i,\lambda_i}^t : \Delta(A_i) \ni \boldsymbol{\pi} \mapsto \langle \boldsymbol{u}_i^t, \boldsymbol{\pi}\rangle - \lambda_i D_{\mathrm{KL}}(\boldsymbol{\pi} \,\|\, \boldsymbol{\tau}_i).
$$

**Observation 1.** *We note the following:*

- *For any $i \in \{1, 2\}$ and $\lambda_i \in \Lambda_i$, the function $\tilde{u}_{i,\lambda_i}^t$ satisfies*

$$
\tilde{u}_{i,\lambda_i}^t(\boldsymbol{\pi}) = \tilde{u}_{i,\lambda_i}^t(\boldsymbol{\pi}') + \langle \nabla\tilde{u}_{i,\lambda_i}^t(\boldsymbol{\pi}'), \boldsymbol{\pi} - \boldsymbol{\pi}'\rangle - \lambda_i D_{\mathrm{KL}}(\boldsymbol{\pi} \,\|\, \boldsymbol{\pi}') \qquad \forall \boldsymbol{\pi}, \boldsymbol{\pi}' \in \Delta(A_i).
$$

- *Furthermore,*

$$-\nabla \tilde{u}_{i,\lambda_i}^t(\boldsymbol{\pi}_{i,\lambda_i}^t) = -\boldsymbol{u}_i^t + \lambda_i \nabla \phi(\boldsymbol{\pi}^t) - \lambda_i \nabla \phi(\boldsymbol{\tau}_i).$$

Using Corollary 1 we have the following

**Lemma 2.** *For any player $i$ and type $\lambda_i \in \Lambda_i$,*

$$\tilde{u}_{i,\lambda_i}^t(\boldsymbol{\pi}) - \tilde{u}_{i,\lambda_i}^t(\boldsymbol{\pi}_{i,\lambda_i}^t) \leq \frac{\|\boldsymbol{u}_i^t\|_\infty^2}{4\lambda_i t + 4/\eta} + \lambda_i \Big( D_{\mathrm{KL}}(\boldsymbol{\pi}_{i,\lambda_i}^t \,\|\, \boldsymbol{\tau}_i) - D_{\mathrm{KL}}(\boldsymbol{\pi}_{i,\lambda_i}^{t+1} \,\|\, \boldsymbol{\tau}_i) \Big)$$
$$- \left( \lambda_i t + \frac{1}{\eta} \right) D_{\mathrm{KL}}(\boldsymbol{\pi} \,\|\, \boldsymbol{\pi}_{i,\lambda_i}^{t+1}) + \left( \lambda_i (t-1) + \frac{1}{\eta} \right) D_{\mathrm{KL}}(\boldsymbol{\pi} \,\|\, \boldsymbol{\pi}_{i,\lambda_i}^t).$$

*Proof.* From Lemma 1,

$$0 = \left( \lambda_i t + \frac{1}{\eta} \right) \Big( - D_{\mathrm{KL}}(\boldsymbol{\pi} \,\|\, \boldsymbol{\pi}_{i,\lambda_i}^{t+1}) + D_{\mathrm{KL}}(\boldsymbol{\pi} \,\|\, \boldsymbol{\pi}_{i,\lambda_i}^t) - D_{\mathrm{KL}}(\boldsymbol{\pi}_{i,\lambda_i}^{t+1} \,\|\, \boldsymbol{\pi}_{i,\lambda_i}^t) \Big) + \langle -\nabla \tilde{u}_{i,\lambda_i}^t(\boldsymbol{\pi}_{i,\lambda_i}^t), \boldsymbol{\pi} - \boldsymbol{\pi}_{i,\lambda_i}^{t+1} \rangle$$

$$= \left( \lambda_i t + \frac{1}{\eta} \right) \Big( - D_{\mathrm{KL}}(\boldsymbol{\pi} \,\|\, \boldsymbol{\pi}_{i,\lambda_i}^{t+1}) + D_{\mathrm{KL}}(\boldsymbol{\pi} \,\|\, \boldsymbol{\pi}_{i,\lambda_i}^t) - D_{\mathrm{KL}}(\boldsymbol{\pi}_{i,\lambda_i}^{t+1} \,\|\, \boldsymbol{\pi}_{i,\lambda_i}^t) \Big)$$
$$+ \langle \nabla \tilde{u}_{i,\lambda_i}^t(\boldsymbol{\pi}_{i,\lambda_i}^t), \boldsymbol{\pi}_{i,\lambda_i}^t + \boldsymbol{\pi}_{i,\lambda_i}^{t+1} \rangle + \langle -\nabla \tilde{u}_{i,\lambda_i}^t(\boldsymbol{\pi}_{i,\lambda_i}^t), \boldsymbol{\pi} - \boldsymbol{\pi}_{i,\lambda_i}^t \rangle$$

$$= \left( \lambda_i t + \frac{1}{\eta} \right) \Big( - D_{\mathrm{KL}}(\boldsymbol{\pi} \,\|\, \boldsymbol{\pi}_{i,\lambda_i}^{t+1}) + D_{\mathrm{KL}}(\boldsymbol{\pi} \,\|\, \boldsymbol{\pi}_{i,\lambda_i}^t) - D_{\mathrm{KL}}(\boldsymbol{\pi}_{i,\lambda_i}^{t+1} \,\|\, \boldsymbol{\pi}_{i,\lambda_i}^t) \Big)$$
$$+ \langle -\nabla \tilde{u}_{i,\lambda_i}^t(\boldsymbol{\pi}_{i,\lambda_i}^t), \boldsymbol{\pi}_{i,\lambda_i}^t - \boldsymbol{\pi}_{i,\lambda_i}^{t+1} \rangle - \tilde{u}_{i,\lambda_i}^t(\boldsymbol{\pi}) + \tilde{u}_{i,\lambda_i}^t(\boldsymbol{\pi}_{i,\lambda_i}^t) - \lambda_i \, D_{\mathrm{KL}}(\boldsymbol{\pi} \,\|\, \boldsymbol{\pi}_{i,\lambda_i}^t).$$

Rearranging, we find

$$\tilde{u}_{i,\lambda_i}^t(\boldsymbol{\pi}) - \tilde{u}_{i,\lambda_i}^t(\boldsymbol{\pi}_{i,\lambda_i}^t) = -\left( \lambda_i t + \frac{1}{\eta} \right) D_{\mathrm{KL}}(\boldsymbol{\pi} \,\|\, \boldsymbol{\pi}_{i,\lambda_i}^{t+1}) + \left( \lambda_i (t-1) + \frac{1}{\eta} \right) D_{\mathrm{KL}}(\boldsymbol{\pi} \,\|\, \boldsymbol{\pi}_{i,\lambda_i}^t)$$
$$- \left( \lambda_i t + \frac{1}{\eta} \right) D_{\mathrm{KL}}(\boldsymbol{\pi}_{i,\lambda_i}^{t+1} \,\|\, \boldsymbol{\pi}_{i,\lambda_i}^t) + \underbrace{\langle -\nabla \tilde{u}_{i,\lambda_i}^t(\boldsymbol{\pi}_{i,\lambda_i}^t), \boldsymbol{\pi}_{i,\lambda_i}^t - \boldsymbol{\pi}_{i,\lambda_i}^{t+1} \rangle}_{(9)}. \tag{10}$$

We now upper bound the term in (9) using convexity of the function $\boldsymbol{\pi} \mapsto D_{\mathrm{KL}}(\boldsymbol{\pi} \,\|\, \boldsymbol{\tau}_i)$, as follows:

$$\langle -\nabla \tilde{u}_{i,\lambda_i}^t(\boldsymbol{\pi}_{i,\lambda_i}^t), \boldsymbol{\pi}_{i,\lambda_i}^t - \boldsymbol{\pi}_{i,\lambda_i}^{t+1} \rangle = \langle -\boldsymbol{u}_i^t, \boldsymbol{\pi}_{i,\lambda_i}^t - \boldsymbol{\pi}_{i,\lambda_i}^{t+1} \rangle + \lambda_i \langle \nabla \phi(\boldsymbol{\pi}_{i,\lambda_i}^t) - \nabla \phi(\boldsymbol{\tau}_i), \boldsymbol{\pi}_{i,\lambda_i}^t - \boldsymbol{\pi}_{i,\lambda_i}^{t+1} \rangle$$
$$\leq \langle -\boldsymbol{u}_i^t, \boldsymbol{\pi}_{i,\lambda_i}^t - \boldsymbol{\pi}_{i,\lambda_i}^{t+1} \rangle + \lambda_i \Big( D_{\mathrm{KL}}(\boldsymbol{\pi}_{i,\lambda_i}^{t+1} \,\|\, \boldsymbol{\tau}_i) - D_{\mathrm{KL}}(\boldsymbol{\pi}_{i,\lambda_i}^t \,\|\, \boldsymbol{\tau}_i) \Big).$$

Substituting the above bound into (10) yields

$$\tilde{u}_{i,\lambda_i}^t(\boldsymbol{\pi}) - \tilde{u}_{i,\lambda_i}^t(\boldsymbol{\pi}_{i,\lambda_i}^t) \leq \langle -\boldsymbol{u}_i^t, \boldsymbol{\pi}_{i,\lambda_i}^t - \boldsymbol{\pi}_{i,\lambda_i}^{t+1} \rangle - \left( \lambda_i t + \frac{1}{\eta} \right) D_{\mathrm{KL}}(\boldsymbol{\pi}_{i,\lambda_i}^{t+1} \,\|\, \boldsymbol{\pi}_{i,\lambda_i}^t)$$
$$+ \lambda_i \Big( D_{\mathrm{KL}}(\boldsymbol{\pi}_{i,\lambda_i}^t \,\|\, \boldsymbol{\tau}_i) - D_{\mathrm{KL}}(\boldsymbol{\pi}_{i,\lambda_i}^{t+1} \,\|\, \boldsymbol{\tau}_i) \Big)$$
$$- \left( \lambda_i t + \frac{1}{\eta} \right) D_{\mathrm{KL}}(\boldsymbol{\pi} \,\|\, \boldsymbol{\pi}_{i,\lambda_i}^{t+1}) + \left( \lambda_i (t-1) + \frac{1}{\eta} \right) D_{\mathrm{KL}}(\boldsymbol{\pi} \,\|\, \boldsymbol{\pi}_{i,\lambda_i}^t)$$
$$\leq \frac{\|\boldsymbol{u}_i^t\|_\infty^2}{4\lambda_i t + 4/\eta} + \left( \lambda_i t + \frac{1}{\eta} \right) \|\boldsymbol{\pi}_{i,\lambda_i}^t - \boldsymbol{\pi}_{i,\lambda_i}^{t+1}\|_1^2 - \left( \lambda_i t + \frac{1}{\eta} \right) D_{\mathrm{KL}}(\boldsymbol{\pi}_{i,\lambda_i}^{t+1} \,\|\, \boldsymbol{\pi}_{i,\lambda_i}^t)$$
$$+ \lambda_i \Big( D_{\mathrm{KL}}(\boldsymbol{\pi}_{i,\lambda_i}^t \,\|\, \boldsymbol{\tau}_i) - D_{\mathrm{KL}}(\boldsymbol{\pi}_{i,\lambda_i}^{t+1} \,\|\, \boldsymbol{\tau}_i) \Big)$$
$$- \left( \lambda_i t + \frac{1}{\eta} \right) D_{\mathrm{KL}}(\boldsymbol{\pi} \,\|\, \boldsymbol{\pi}_{i,\lambda_i}^{t+1}) + \left( \lambda_i (t-1) + \frac{1}{\eta} \right) D_{\mathrm{KL}}(\boldsymbol{\pi} \,\|\, \boldsymbol{\pi}_{i,\lambda_i}^t),$$

where the second inequality follows from Young's inequality. Finally, by using the strong convexity of the KL divergence between points $\boldsymbol{\pi}_{i,\lambda_i}^t$ and $\boldsymbol{\pi}_{i,\lambda_i}^{t+1}$, that is,

$$D_{\mathrm{KL}}(\boldsymbol{\pi}_{i,\lambda_i}^{t+1} \,\|\, \boldsymbol{\pi}_{i,\lambda_i}^t) \geq \|\boldsymbol{\pi}_{i,\lambda_i}^{t+1} - \boldsymbol{\pi}_{i,\lambda_i}^t\|_1^2,$$

yields the statement. □

Noting that the right-hand side of Lemma 2 is telescopic, we immediately have the following.

**Theorem 3.** *For any player $i$ and type $\lambda_i \in \Lambda_i$, and policy $\boldsymbol{\pi} \in \Delta(A_i)$, the following regret bound holds at all times $T$:*

$$\sum_{t=1}^{T} \tilde{u}_{i,\lambda_i}^t(\boldsymbol{\pi}) - \tilde{u}_{i,\lambda_i}^t(\boldsymbol{\pi}_{i,\lambda_i}^t) \leq \frac{W^2}{4} \min\left\{\frac{2\log T}{\lambda_i}, T\eta\right\} + \frac{\log n_i}{\eta} + \lambda_i(\log n_i + Q_i).$$

*Proof.* From Lemma 2 we have that

$$\sum_{t=1}^{T} \tilde{u}_{i,\lambda_i}^t(\boldsymbol{\pi}) - \tilde{u}_{i,\lambda_i}^t(\boldsymbol{\pi}_{i,\lambda_i}^t) \leq \left(\frac{W^2}{4} \sum_{t=1}^{T} \frac{1}{\lambda_i t + 1/\eta}\right) + \lambda_i D_{\mathrm{KL}}(\boldsymbol{\pi}_{i,\lambda_i}^1 \| \boldsymbol{\tau}_i) + \frac{D_{\mathrm{KL}}(\boldsymbol{\pi} \| \boldsymbol{\pi}_{i,\lambda_i}^1)}{\eta}$$

$$\leq \frac{W^2}{4}\left(\sum_{t=1}^{T} \min\left\{\frac{1}{\lambda_i t}, \eta\right\}\right) + \lambda_i(\log n_i + Q_i) + \frac{\log n_i}{\eta}$$

$$\leq \frac{W^2}{4} \min\left\{\frac{2\log T}{\lambda_i}, \eta T\right\} + \lambda_i(\log n_i + Q_i) + \frac{\log n_i}{\eta},$$

where the second inequality follows from the fact that $\lambda_i t + 1/\eta \geq \max\{\lambda_i t, 1/\eta\}$ and the fact that $\boldsymbol{\pi}_{i,\lambda_i}^1$ is the uniform strategy. $\qquad\square$

### D.2 LAST-ITERATE CONVERGENCE IN TWO-PLAYER ZERO-SUM GAMES

In two-player game with payoff matrix $\boldsymbol{A}$ for Player 1, a Bayes-Nash equilibrium to the regularized game is a collection of policies $(\boldsymbol{\pi}_{i,\lambda_i}^*)$ such that for any supported type $\lambda_i$ of Player $i \in \{1, 2\}$, the policy $\boldsymbol{\pi}_{i,\lambda_i}^*$ is a best response to the average policy of the opponent. In symbols,

$$\boldsymbol{\pi}_{1,\lambda_1}^* \in \underset{\boldsymbol{\pi} \in \Delta(A_1)}{\arg\max}\left\{\langle \boldsymbol{A} \underset{\lambda_2 \sim \beta_2}{\mathbb{E}}[\boldsymbol{\pi}_{2,\lambda_2}^*], \boldsymbol{\pi}\rangle + \lambda_1 D_{\mathrm{KL}}(\boldsymbol{\pi} \| \boldsymbol{\tau}_1)\right\} \qquad \forall \lambda_1 \in \Lambda_1,$$

$$\boldsymbol{\pi}_{2,\lambda_2}^* \in \underset{\boldsymbol{\pi} \in \Delta(A_2)}{\arg\max}\left\{\langle -\boldsymbol{A}^\top \underset{\lambda_1 \sim \beta_1}{\mathbb{E}}[\boldsymbol{\pi}_{1,\lambda_1}^*], \boldsymbol{\pi}\rangle + \lambda_2 D_{\mathrm{KL}}(\boldsymbol{\pi} \| \boldsymbol{\tau}_2)\right\} \qquad \forall \lambda_2 \in \Lambda_2.$$

Denoting $\bar{\boldsymbol{\pi}}_1^* := \mathbb{E}_{\lambda_1 \sim \beta_1}[\boldsymbol{\pi}_{1,\lambda_1}^*], \bar{\boldsymbol{\pi}}_2^* := \mathbb{E}_{\lambda_2 \sim \beta_2}[\boldsymbol{\pi}_{2,\lambda_2}^*]$, the first-order optimality conditions for the best response problems above are

$$\langle \boldsymbol{A}\,\bar{\boldsymbol{\pi}}_2^* + \lambda_1 \nabla\phi(\boldsymbol{\pi}_{1,\lambda_1}^*) - \lambda_1 \nabla\phi(\boldsymbol{\tau}_1), \boldsymbol{\pi}_{1,\lambda_1}^* - \boldsymbol{\pi}_{1,\lambda_1}'\rangle \geq 0 \qquad \forall \boldsymbol{\pi}_{1,\lambda_1}' \in \Delta(A_1),$$

$$\langle -\boldsymbol{A}^\top \bar{\boldsymbol{\pi}}_1^* + \lambda_2 \nabla\phi(\boldsymbol{\pi}_{2,\lambda_2}^*) - \lambda_2 \nabla\phi(\boldsymbol{\tau}_2), \boldsymbol{\pi}_{2,\lambda_2}^* - \boldsymbol{\pi}_{2,\lambda_2}'\rangle \geq 0 \qquad \forall \boldsymbol{\pi}_{2,\lambda_2}' \in \Delta(A_2).$$

We also mention the following standard lemma.

**Lemma 3.** *Let $(\boldsymbol{\pi}_{i,\lambda_i}^*)_{i\in\{1,2\},\lambda_1\in\Lambda_i}$ be the unique Bayes-Nash equilibrium of the regularized game. Let policies $\boldsymbol{\pi}_{i,\lambda_i}'$ be arbitrary, and let:*

- $\bar{\boldsymbol{\pi}}_1' := \mathbb{E}_{\lambda_1 \sim \beta_1}[\boldsymbol{\pi}_{1,\lambda_1}'], \quad \bar{\boldsymbol{\pi}}_2' := \mathbb{E}_{\lambda_2 \sim \beta_2}[\boldsymbol{\pi}_{2,\lambda_2}'];$

- $\alpha := \underset{\lambda_1 \sim \beta_1}{\mathbb{E}}[\langle -\boldsymbol{A}\bar{\boldsymbol{\pi}}_2' + \lambda_1 \nabla\phi(\boldsymbol{\pi}_{1,\lambda_1}') - \lambda_1 \nabla\phi(\boldsymbol{\tau}_1), \boldsymbol{\pi}_{1,\lambda_1}^* - \boldsymbol{\pi}_{1,\lambda_1}'\rangle];$

- $\beta := \underset{\lambda_2 \sim \beta_2}{\mathbb{E}}[\langle \boldsymbol{A}^\top \bar{\boldsymbol{\pi}}_1' + \lambda_2 \nabla\phi(\boldsymbol{\pi}_{2,\lambda_2}') - \lambda_2 \nabla\phi(\boldsymbol{\tau}_2), \boldsymbol{\pi}_{2,\lambda_2}^* - \boldsymbol{\pi}_{2,\lambda_2}'\rangle].$

*Then,*

$$\alpha + \beta \leq -\sum_{i\in\{1,2\}} \underset{\lambda_i \sim \beta_i}{\mathbb{E}}[\lambda_i\, D_{\mathrm{KL}}(\boldsymbol{\pi}_{i,\lambda_i}' \| \boldsymbol{\pi}_{i,\lambda_i}^*) + \lambda_i\, D_{\mathrm{KL}}(\boldsymbol{\pi}_{i,\lambda_i}^* \| \boldsymbol{\pi}_{i,\lambda_i}')].$$

The following potential function will be key in the analysis:

$$\Psi^t := \sum_{i\in\{1,2\}} \underset{\lambda_i \sim \beta_i}{\mathbb{E}}\left[\left(\lambda_i(t-1) + \frac{1}{\eta}\right)D_{\mathrm{KL}}(\boldsymbol{\pi}_{i,\lambda_i}^* \| \boldsymbol{\pi}_{i,\lambda_i}^t) + \lambda_i D_{\mathrm{KL}}(\boldsymbol{\pi}_{i,\lambda_i}^t \| \boldsymbol{\tau}_i)\right], \quad t \in \{1, 2, \dots\}.$$

**Proposition 1.** *At all times $t \in \{1, 2, \dots\}$, let*

$$\bar{\boldsymbol{\pi}}_{-i}^t := \underset{\lambda_{-i} \sim \beta_{-i}}{\mathbb{E}} \left[ \boldsymbol{\pi}_{-i,\lambda_{-i}}^t \right].$$

*The potential $\Psi^t$ satisfies the inequality*

$$\Psi^{t+1} \leq \Psi^t + \sum_{i \in \{1,2\}} \underset{\lambda_i \sim \beta_i}{\mathbb{E}} \left[ \frac{\|\boldsymbol{u}_i^t\|_\infty^2}{4\lambda_i t + 4/\eta} + \left\langle \boldsymbol{A}_i \bar{\boldsymbol{\pi}}_{-i}^t - \boldsymbol{u}_i^t, \boldsymbol{\pi}_{i,\lambda_i}^* - \boldsymbol{\pi}_{i,\lambda_i}^t \right\rangle \right].$$

*Proof.* By multiplying both sides of Corollary 1 for the choice $\boldsymbol{\pi} = \boldsymbol{\pi}_{i,\lambda_i}^*$, taking expectations over $\lambda_i \sim \beta_i$, and summing over the player $i \in \{1, 2\}$, we find

$$\sum_{i \in \{1,2\}} \underset{\lambda_i \sim \beta_i}{\mathbb{E}} \left[ \left(\lambda_i t + \frac{1}{\eta}\right) D_{\mathrm{KL}}(\boldsymbol{\pi}_{i,\lambda_i}^* \,\|\, \boldsymbol{\pi}_{i,\lambda_i}^{t+1}) \right] = \sum_{i \in \{1,2\}} \underset{\lambda_i \sim \beta_i}{\mathbb{E}} \left[ \left(\lambda_i t + \frac{1}{\eta}\right) D_{\mathrm{KL}}(\boldsymbol{\pi}_{i,\lambda_i}^* \,\|\, \boldsymbol{\pi}_{i,\lambda_i}^t) \right]$$

$$- \sum_{i \in \{1,2\}} \underset{\lambda_i \sim \beta_i}{\mathbb{E}} \left[ \left(\lambda_i t + \frac{1}{\eta}\right) D_{\mathrm{KL}}(\boldsymbol{\pi}_{i,\lambda_i}^{t+1} \,\|\, \boldsymbol{\pi}_{i,\lambda_i}^t) \right]$$

$$+ \underbrace{\sum_{i \in \{1,2\}} \underset{\lambda_i \sim \beta_i}{\mathbb{E}} \left[ \left\langle -\boldsymbol{u}_i^t + \lambda_i \nabla\phi(\boldsymbol{\pi}_{i,\lambda_i}^t) - \lambda_i \nabla\phi(\boldsymbol{\tau}_i), \boldsymbol{\pi}_{i,\lambda_i}^* - \boldsymbol{\pi}_{i,\lambda_i}^{t+1} \right\rangle \right]}_{(\clubsuit)}. \quad (11)$$

We now proceed to analyze the last summation on the right-hand side. First,

$$(\clubsuit) = \underbrace{\sum_{i \in \{1,2\}} \underset{\lambda_i \sim \beta_i}{\mathbb{E}} \left[ \left\langle -\boldsymbol{A}_i \bar{\boldsymbol{\pi}}_{-i}^t + \lambda_i \nabla\phi(\boldsymbol{\pi}_{i,\lambda_i}^t) - \lambda_i \nabla\phi(\boldsymbol{\tau}_i), \boldsymbol{\pi}_{i,\lambda_i}^* - \boldsymbol{\pi}_{i,\lambda_i}^t \right\rangle \right]}_{(\spadesuit)}$$

$$+ \underbrace{\sum_{i \in \{1,2\}} \underset{\lambda_i \sim \beta_i}{\mathbb{E}} \left[ \left\langle -\boldsymbol{u}_i^t + \lambda_i \nabla\phi(\boldsymbol{\pi}_{i,\lambda_i}^t) - \lambda_i \nabla\phi(\boldsymbol{\tau}_i), \boldsymbol{\pi}_{i,\lambda_i}^t - \boldsymbol{\pi}_{i,\lambda_i}^{t+1} \right\rangle \right]}_{(\heartsuit)}$$

$$+ \sum_{i \in \{1,2\}} \underset{\lambda_i \sim \beta_i}{\mathbb{E}} \left[ \left\langle \boldsymbol{A}_i \bar{\boldsymbol{\pi}}_{-i}^t - \boldsymbol{u}_i^t, \boldsymbol{\pi}_{i,\lambda_i}^* - \boldsymbol{\pi}_{i,\lambda_i}^t \right\rangle \right]. \quad (12)$$

Using Lemma 3 we can immediately write

$$(\spadesuit) \leq \sum_{i \in \{1,2\}} \underset{\lambda_i \sim \beta_i}{\mathbb{E}} \left[ -\lambda_i D_{\mathrm{KL}}(\boldsymbol{\pi}_{i,\lambda_i}^* \,\|\, \boldsymbol{\pi}_{i,\lambda_i}^t) \right].$$

By manipulating the inner product in $(\heartsuit)$, we have

$$(\heartsuit) = \sum_{i \in \{1,2\}} \underset{\lambda_i \sim \beta_i}{\mathbb{E}} \left[ \langle -\boldsymbol{u}_i^t, \boldsymbol{\pi}_{i,\lambda_i}^t - \boldsymbol{\pi}_{i,\lambda_i}^{t+1} \rangle - \lambda_i \left\langle \nabla\phi(\boldsymbol{\pi}_{i,\lambda_i}^t) - \phi(\boldsymbol{\pi}_{i,\lambda_i}^{t+1}), \boldsymbol{\pi}_{i,\lambda_i}^{t+1} - \boldsymbol{\pi}_{i,\lambda_i}^t \right\rangle \right]$$

$$\leq \sum_{i \in \{1,2\}} \underset{\lambda_i \sim \beta_i}{\mathbb{E}} \left[ \langle -\boldsymbol{u}_i^t, \boldsymbol{\pi}_{i,\lambda_i}^t - \boldsymbol{\pi}_{i,\lambda_i}^{t+1} \rangle + \lambda_i \left( D_{\mathrm{KL}}(\boldsymbol{\pi}_{i,\lambda_i}^{t+1} \,\|\, \boldsymbol{\tau}_i) - D_{\mathrm{KL}}(\boldsymbol{\pi}_{i,\lambda_i}^t \,\|\, \boldsymbol{\tau}_i) \right) \right]$$

$$\leq \sum_{i \in \{1,2\}} \underset{\lambda_i \sim \beta_i}{\mathbb{E}} \left[ \frac{\|\boldsymbol{u}_i^t\|_\infty^2}{4\lambda_i t + 4/\eta} + \left(\lambda_i t + \frac{1}{\eta}\right) \left\|\boldsymbol{\pi}_{i,\lambda_i}^t - \boldsymbol{\pi}_{i,\lambda_i}^{t+1}\right\|_1^2 \right]$$

$$+ \sum_{i \in \{1,2\}} \underset{\lambda_i \sim \beta_i}{\mathbb{E}} \left[ \lambda_i \left( D_{\mathrm{KL}}(\boldsymbol{\pi}_{i,\lambda_i}^{t+1} \,\|\, \boldsymbol{\tau}_i) - D_{\mathrm{KL}}(\boldsymbol{\pi}_{i,\lambda_i}^t \,\|\, \boldsymbol{\tau}_i) \right) \right],$$

where the last inequality follow from the fact that $ab \leq a^2/(4\rho) + \rho b^2$ for all choices of $a, b \geq 0$ and $\rho > 0$. Substituting the individual bounds into (12) yields

$$(\clubsuit) \leq \sum_{i \in \{1,2\}} \underset{\lambda_i \sim \beta_i}{\mathbb{E}} \left[ \lambda_i \left( D_{\mathrm{KL}}(\boldsymbol{\pi}_{i,\lambda_i}^{t+1} \,\|\, \boldsymbol{\tau}_i) - D_{\mathrm{KL}}(\boldsymbol{\pi}_{i,\lambda_i}^t \,\|\, \boldsymbol{\tau}_i) \right) \right]$$

$$+ \sum_{i \in \{1,2\}} \mathop{\mathbb{E}}_{\lambda_i \sim \beta_i} \left[ \frac{\|\boldsymbol{u}_i^t\|_\infty^2}{4\lambda_i t + 4/\eta} + \left( \lambda_i t + \frac{1}{\eta} \right) \left\| \boldsymbol{\pi}_{i,\lambda_i}^t - \boldsymbol{\pi}_{i,\lambda_i}^{t+1} \right\|_1^2 \right]$$

$$+ \sum_{i \in \{1,2\}} \mathop{\mathbb{E}}_{\lambda_i \sim \beta_i} \left[ \langle \boldsymbol{A}_i \bar{\boldsymbol{\pi}}_{-i}^t - \boldsymbol{u}_i^t, \boldsymbol{\pi}_{i,\lambda_i}^* - \boldsymbol{\pi}_{i,\lambda_i}^t \rangle \right].$$

Finally, plugging the above bound into (11) and rearranging terms yields

$$\Psi^{t+1} \le \Psi^t + \sum_{i \in \{1,2\}} \mathop{\mathbb{E}}_{\lambda_i \sim \beta_i} \left[ -\left( \lambda_i t + \frac{1}{\eta} \right) D_{\mathrm{KL}}(\boldsymbol{\pi}_{i,\lambda_i}^{t+1} \| \boldsymbol{\pi}_{i,\lambda_i}^t) \right]$$

$$+ \sum_{i \in \{1,2\}} \mathop{\mathbb{E}}_{\lambda_i \sim \beta_i} \left[ \frac{\|\boldsymbol{u}_i^t\|_\infty^2}{4\lambda_i t + 4/\eta} + \left( \lambda_i t + \frac{1}{\eta} \right) \left\| \boldsymbol{\pi}_{i,\lambda_i}^t - \boldsymbol{\pi}_{i,\lambda_i}^{t+1} \right\|_1^2 \right]$$

$$+ \sum_{i \in \{1,2\}} \mathop{\mathbb{E}}_{\lambda_i \sim \beta_i} \left[ \langle \boldsymbol{A}_i \bar{\boldsymbol{\pi}}_{-i}^t - \boldsymbol{u}_i^t, \boldsymbol{\pi}_{i,\lambda_i}^* - \boldsymbol{\pi}_{i,\lambda_i}^t \rangle \right].$$

$$\le \Psi^t + \sum_{i \in \{1,2\}} \mathop{\mathbb{E}}_{\lambda_i \sim \beta_i} \left[ -\left( \lambda_i t + \frac{1}{\eta} \right) \left\| \boldsymbol{\pi}_{i,\lambda_i}^{t+1} - \boldsymbol{\pi}_{i,\lambda_i}^t \right\|_1^2 \right]$$

$$+ \sum_{i \in \{1,2\}} \mathop{\mathbb{E}}_{\lambda_i \sim \beta_i} \left[ \frac{\|\boldsymbol{u}_i^t\|_\infty^2}{4\lambda_i t + 4/\eta} + \left( \lambda_i t + \frac{1}{\eta} \right) \left\| \boldsymbol{\pi}_{i,\lambda_i}^t - \boldsymbol{\pi}_{i,\lambda_i}^{t+1} \right\|_1^2 \right]$$

$$+ \sum_{i \in \{1,2\}} \mathop{\mathbb{E}}_{\lambda_i \sim \beta_i} \left[ \langle \boldsymbol{A}_i \bar{\boldsymbol{\pi}}_{-i}^t - \boldsymbol{u}_i^t, \boldsymbol{\pi}_{i,\lambda_i}^* - \boldsymbol{\pi}_{i,\lambda_i}^t \rangle \right].$$

$$\le \Psi^t + \sum_{i \in \{1,2\}} \mathop{\mathbb{E}}_{\lambda_i \sim \beta_i} \left[ \frac{\|\boldsymbol{u}_i^t\|_\infty^2}{4\lambda_i t + 4/\eta} + \langle \boldsymbol{A}_i \bar{\boldsymbol{\pi}}_{-i}^t - \boldsymbol{u}_i^t, \boldsymbol{\pi}_{i,\lambda_i}^* - \boldsymbol{\pi}_{i,\lambda_i}^t \rangle \right],$$

as we wanted to show. □

**Theorem 4.** *As in Proposition 1, let*

$$\bar{\boldsymbol{\pi}}_{-i}^t := \mathop{\mathbb{E}}_{\lambda_{-i} \sim \beta_{-i}} \left[ \boldsymbol{\pi}_{-i,\lambda_{-i}}^t \right].$$

*Let $\tilde{D}_{\mathrm{KL}}^T$ be the notion of distance defined as*

$$\tilde{D}_{\mathrm{KL}}^T := \sum_{i \in \{1,2\}} \mathop{\mathbb{E}}_{\lambda_i \sim \beta_i} \left[ (\lambda_i + \kappa_{T-1}) D_{\mathrm{KL}}(\boldsymbol{\pi}_{i,\lambda_i}^* \| \boldsymbol{\pi}_{i,\lambda_i}^T) \right].$$

*At all times $T = 2, 3, \ldots,$*

$$\tilde{D}_{\mathrm{KL}}^T \le \frac{1}{T} \left( \rho + \frac{\log n_i}{\eta} + \frac{W^2}{2} \sum_{i \in \{1,2\}} \mathop{\mathbb{E}}_{\lambda_i \sim \beta_i} \left[ \min \left\{ \frac{2 \log T}{\lambda_i}, \eta T \right\} \right] \right)$$

$$+ \frac{2}{T} \sum_{t=1}^T \sum_{i \in \{1,2\}} \mathop{\mathbb{E}}_{\lambda_i \sim \beta_i} \left[ \langle \boldsymbol{A}_i \bar{\boldsymbol{\pi}}_{-i}^t - \boldsymbol{u}_i^t, \boldsymbol{\pi}_{i,\lambda_i}^* - \boldsymbol{\pi}_{i,\lambda_i}^t \rangle \right],$$

*where*

$$\rho := 2 \sum_{i \in \{1,2\}} \mathop{\mathbb{E}}_{\lambda_i \sim \beta_i} [\lambda_i] (\log n_i + Q_i).$$

*Proof.* Using the bound on $\Psi^{t+1} - \Psi^t$ given by Proposition 1 we obtain

$$\Psi^T - \Psi^1 = \sum_{t=1}^{T-1} (\Psi^{t+1} - \Psi^t)$$

$$\leq \sum_{t=1}^{T-1} \sum_{i \in \{1,2\}} \mathbb{E}_{\lambda_i \sim \beta_i} \left[ \frac{\|\boldsymbol{u}_i^t\|_\infty^2}{4\lambda_i t + 4/\eta} + \left\langle \boldsymbol{A}_i \bar{\boldsymbol{\pi}}_{-i}^t - \boldsymbol{u}_i^t, \boldsymbol{\pi}_{i,\lambda_i}^* - \boldsymbol{\pi}_{i,\lambda_i}^t \right\rangle \right]$$

$$= \frac{1}{4} \sum_{i \in \{1,2\}} \mathbb{E}_{\lambda_i \sim \beta_i} \left[ \sum_{t=1}^{T} \frac{\|\boldsymbol{u}_i^t\|_\infty^2}{\lambda_i t + 1/\eta} \right] + \sum_{t=1}^{T} \sum_{i \in \{1,2\}} \mathbb{E}_{\lambda_i \sim \beta_i} \left[ \left\langle \boldsymbol{A}_i \bar{\boldsymbol{\pi}}_{-i}^t - \boldsymbol{u}_i^t, \boldsymbol{\pi}_{i,\lambda_i}^* - \boldsymbol{\pi}_{i,\lambda_i}^t \right\rangle \right]$$

$$\leq \frac{1}{4} \sum_{i \in \{1,2\}} \mathbb{E}_{\lambda_i \sim \beta_i} \left[ \sum_{t=1}^{T-1} \frac{W^2}{\lambda_i t + 1/\eta} \right] + \sum_{t=1}^{T} \sum_{i \in \{1,2\}} \mathbb{E}_{\lambda_i \sim \beta_i} \left[ \left\langle \boldsymbol{A}_i \bar{\boldsymbol{\pi}}_{-i}^t - \boldsymbol{u}_i^t, \boldsymbol{\pi}_{i,\lambda_i}^* - \boldsymbol{\pi}_{i,\lambda_i}^t \right\rangle \right].$$

We can now bound

$$\sum_{t=1}^{T} \frac{W^2}{\lambda_i t + 1/\eta} \leq W^2 \sum_{t=1}^{T} \min \left\{ \frac{1}{\lambda_i t}, \eta \right\}$$

$$\leq W^2 \min \left\{ \sum_{t=1}^{T} \frac{1}{\lambda_i t}, \sum_{t=1}^{T} \eta \right\}$$

$$\leq W^2 \min \left\{ \frac{2 \log T}{\lambda_i}, T\eta \right\}.$$

On the other hand, note that

$$\Psi^T - \Psi^1 = -\Psi^1 + \sum_{i \in \{1,2\}} \mathbb{E}_{\lambda_i \sim \beta_i} \left[ \left( \lambda_i (T-1) + \frac{1}{\eta} \right) D_{\mathrm{KL}}(\boldsymbol{\pi}_{i,\lambda_i}^* \,\|\, \boldsymbol{\pi}_{i,\lambda_i}^T) + \lambda_i D_{\mathrm{KL}}(\boldsymbol{\pi}_{i,\lambda_i}^T \,\|\, \boldsymbol{\tau}_i) \right]$$

$$\geq -\Psi^1 + \sum_{i \in \{1,2\}} (T-1) \mathbb{E}_{\lambda_i \sim \beta_i} \left[ (\lambda_i + \kappa_{T-1}) D_{\mathrm{KL}}(\boldsymbol{\pi}_{i,\lambda_i}^* \,\|\, \boldsymbol{\pi}_{i,\lambda_i}^T) \right]$$

$$= (T-1)\tilde{D}_{\mathrm{KL}}^T - \sum_{i \in \{1,2\}} \mathbb{E}_{\lambda_i \sim \beta_i} \left[ \frac{D_{\mathrm{KL}}(\boldsymbol{\pi}_{i,\lambda_i}^* \,\|\, \boldsymbol{\pi}_{i,\lambda_i}^1)}{\eta} - \lambda_i D_{\mathrm{KL}}(\boldsymbol{\pi}_{i,\lambda_i}^1 \,\|\, \boldsymbol{\tau}_i) \right]$$

$$\geq (T-1)\tilde{D}_{\mathrm{KL}}^T - \sum_{i \in \{1,2\}} \mathbb{E}_{\lambda_i \sim \beta_i} \left[ \frac{\log n_i}{\eta} + \lambda_i (\log n_i + Q_i) \right]$$

$$= (T-1)\tilde{D}_{\mathrm{KL}}^T - \rho,$$

where the last inequality follows from expanding the definition of the KL divergence and using the fact that $\boldsymbol{\pi}_{i,\lambda_i}^1$ is the uniform strategy. Combining the inequalities and dividing by $T-1$ yields

$$\tilde{D}_{\mathrm{KL}}^T \leq \frac{W^2}{4} \sum_{i \in \{1,2\}} \mathbb{E}_{\lambda_i \sim \beta_i} \left[ \min \left\{ \frac{2 \log T}{(T-1)\lambda_i}, \frac{T}{T-1}\eta \right\} \right] + \frac{\rho}{T-1}$$

$$+ \frac{1}{T-1} \sum_{t=1}^{T} \sum_{i \in \{1,2\}} \mathbb{E}_{\lambda_i \sim \beta_i} \left[ \left\langle \boldsymbol{A}_i \bar{\boldsymbol{\pi}}_{-i}^t - \boldsymbol{u}_i^t, \boldsymbol{\pi}_{i,\lambda_i}^* - \boldsymbol{\pi}_{i,\lambda_i}^t \right\rangle \right].$$

Finally, using the fact that $2(T-1) \geq T$ yields the statement. $\qquad \square$

**Theorem 5** (Last-iterate convergence of DiL-piKL in two-player zero-sum games). *Let $\rho$ be as in the statement of Theorem 4. When both players in a zero-sum game learn using DiL-piKL for $T$ iterations, their policies converge to the unique Bayes-Nash equilibrium $(\boldsymbol{\pi}_1^*, \boldsymbol{\pi}_2^*)$ of the regularized game defined by utilities* (4)*, in the following senses:*

*(a) In expectation: for all $i \in \{1, 2\}$ and $\lambda_i \in \Lambda_i$, at a rate of roughly $\log T/(\lambda_i T)$*

$$\mathbb{E} \left[ D_{\mathrm{KL}}(\boldsymbol{\pi}_{i,\lambda_i}^* \,\|\, \boldsymbol{\pi}_{i,\lambda_i}^T) \right] \leq \frac{1}{\lambda_i T} \left( \rho + \frac{\log n_i}{\eta} + \frac{W^2}{2} \sum_{j \in \{1,2\}} \mathbb{E}_{\lambda_j \sim \beta_j} \left[ \min \left\{ \frac{2 \log T}{\lambda_j}, \eta T \right\} \right] \right).$$

*(We remark that for $\eta = 1/\sqrt{T}$ the convergence is never slower than $1/\sqrt{T}$).*

*(b)* *With high probability, at a rate of roughly $1/\sqrt{T}$: for any $\delta \in (0, 1)$ and Player $i \in \{1, 2\}$,*

$$\mathbb{P}\left[\forall \lambda_i \in \Lambda_i : D_{\mathrm{KL}}(\boldsymbol{\pi}^*_{i,\lambda_i} \,\|\, \boldsymbol{\pi}^T_{i,\lambda_i}) \leq \mathbb{E}\left[D_{\mathrm{KL}}(\boldsymbol{\pi}^*_{i,\lambda_i} \,\|\, \boldsymbol{\pi}^T_{i,\lambda_i})\right] + \frac{8\sqrt{2}\,W}{\lambda_i\sqrt{T}}\sqrt{\log\frac{|\Lambda_i|}{\delta}}\right] \geq 1 - \delta.$$

*A n upper bound on $\mathbb{E}\left[D_{\mathrm{KL}}(\boldsymbol{\pi}^*_{i,\lambda_i} \,\|\, \boldsymbol{\pi}^T_{i,\lambda_i})\right]$ was given in the previous point.*

*(c)* *Almost surely in the limit:*

$$\mathbb{P}\left[\forall \lambda_i \in \Lambda_i : D_{\mathrm{KL}}(\boldsymbol{\pi}^*_{i,\lambda_i} \,\|\, \boldsymbol{\pi}^T_{i,\lambda_i}) \xrightarrow{T \to +\infty} 0\right] = 1 \qquad \forall i \in \{1, 2\}.$$

*Proof.* We prove the three statements incrementally.

(a) Let $\mathcal{F}_t$ be the $\sigma$-algebra generated by $\{\boldsymbol{u}_i^{t'} \mid t' = 1, \ldots, t-1, i \in \{1, 2\}\}$. We let $\mathbb{E}_t[\,\cdot\,] := \mathbb{E}[\,\cdot\,\mid\,\mathcal{F}_t]$. Since piKL is a deterministic algorithm, $\boldsymbol{\pi}_{i,\lambda_i}^t$ is $\mathcal{F}_t$-measurable. Hence, given that $\boldsymbol{u}_i^t$ is an unbiased estimator of $\boldsymbol{A}_i\bar{\boldsymbol{\pi}}_{-i}^t$ we have that at all times $t$

$$\mathbb{E}_t\left[\langle \boldsymbol{A}_i\bar{\boldsymbol{\pi}}_{-i}^t - \boldsymbol{u}_i^t, \boldsymbol{\pi}_{i,\lambda_i}^* - \boldsymbol{\pi}_{i,\lambda_i}^t\rangle\right] = \langle \mathbb{E}_t[\boldsymbol{A}_i\bar{\boldsymbol{\pi}}_{-i}^t - \boldsymbol{u}_i^t], \boldsymbol{\pi}_{i,\lambda_i}^* - \boldsymbol{\pi}_{i,\lambda_i}^t\rangle = 0. \tag{13}$$

Note that from the definition of $\tilde{D}_{\mathrm{KL}}^T$ given in Theorem 4

$$D_{\mathrm{KL}}(\boldsymbol{\pi}^*_{i,\lambda_i} \,\|\, \boldsymbol{\pi}^T_{i,\lambda_i}) \leq \frac{1}{\lambda_i}\tilde{D}_{\mathrm{KL}}^T. \tag{14}$$

Hence, taking expectations and using (13) yields the statement.

(b) To prove high-probability convergence, we use the Azuma-Hoeffding concentration inequality. In particular, (13) shows that the stochastic process

$$\left(\sum_{j \in \{1,2\}} \mathbb{E}_{\lambda_j \sim \beta_j}\left[\langle \boldsymbol{A}_j\bar{\boldsymbol{\pi}}_{-j}^t - \boldsymbol{u}_j^t, \boldsymbol{\pi}_j^* - \boldsymbol{\pi}_j^t\rangle\right]\right)_{t=1,2,\ldots}$$

is a martingale difference sequence adapted to the filtration $\mathcal{F}_t$. Furthermore, note that

$$\left|\sum_{j \in \{1,2\}} \mathbb{E}_{\lambda_j \sim \beta_j}\left[\langle \boldsymbol{A}_j\bar{\boldsymbol{\pi}}_{-j}^t - \boldsymbol{u}_j^t, \boldsymbol{\pi}_{j,\lambda_j}^* - \boldsymbol{\pi}_{j,\lambda_j}^t\rangle\right]\right| \leq 4W$$

for all $t$. Hence, using the Azuma-Hoeffding inequality for martingale difference sequences we obtain that for all $\delta \in (0, 1)$,

$$\mathbb{P}\left[\sum_{t=1}^T \sum_{j \in \{1,2\}} \mathbb{E}_{\lambda_j \sim \beta_j}\left[\langle \boldsymbol{A}_j\bar{\boldsymbol{\pi}}_{-j}^t - \boldsymbol{u}_j^t, \boldsymbol{\pi}_j^* - \boldsymbol{\pi}_j^t\rangle\right] \leq 4W\sqrt{2T\log\frac{1}{\delta}}\right] \geq 1 - \delta.$$

Plugging the above probability bound in the statement of Theorem 4 and using the union bound over $\lambda_i \in \Lambda_i$ yields the statement.

(c) follows from (b) via a standard application of the Borel-Cantelli lemma.

$\square$

# E  MODEL ARCHITECTURE

Our model architecture closely resembles the architecture used in past work on no-press Diplomacy (Bakhtin et al., 2021; Jacob et al., 2022; Anthony et al., 2020; Paquette et al., 2019; Gray et al., 2020).

Given a gamestate, to construct the input to the model, for each of the 81 possible locations and/or special coastal areas on the board that a unit can occupy, we encode the 38 feature channels described

| Feature | Type | Number of Channels |
|---|---|---:|
| Presence of army/fleet? | Binary | 2 |
| Army/fleet owner | One-hot (7 players), or all zero | 7 |
| Build turn build/disband | Binary | 2 |
| Dislodged army/fleet? | Binary | 2 |
| Dislodged unit owner | One-hot (7 players), or all zero | 7 |
| Land/coast/water | One-hot | 3 |
| Supply center owner | One-hot (7 players), or all zero | 8 |
| Home center | One-hot (7 players), or all zero | 7 |

Table 3: Per-location board state input features

| Feature | Type | Number of Channels |
|---|---|---:|
| Number of builds allowed during winter | Float | 1 |

Table 4: Per-player board state input features

in Table 3 for that location. We also encode the previous board state in this way, as well using an encoding of the order history as described in Gray et al. (2020) provides an additional 202 channels per board location indicating the prior orders at that location.

Separately, we also encode per-player and global features of the gamestate into additional tensors (Table 4,Table 5). Each of these tensors (per-location, per-player, global) is passed through a linear layer with 224 output channels, and then all three are concatenated to a single (81+7+1) x 224 tensor. Thereafter, following Bakhtin et al. (2021), we apply a learnable positional bias and pass the result to a to a standard transformer encoder architecture with 10 layers, channel width 224, 8 dot-product-self-attention heads per layer, and GeLU activation.

Finally we decode the policy via the same LSTM decoder head as Gray et al. (2020), and predict the game values of all 7 players using a value head that also follows Gray et al. (2020), applying a fully-connected linear layer with 224 output channels, ReLU activation, a linear layer with 7 channels, and a softmax. See Figure 4 for a graphical diagram of the model. For baselines BRBot and HedgeBot in Table 1, we also use a slightly improved value head described in FAIR et al. (2022).

## F  HUMAN IMITATION ANCHOR POLICY TRAINING

Similar to prior work (Bakhtin et al., 2021; Jacob et al., 2022; Gray et al., 2020), to obtain a human imitation anchor policy with which to use for piKL regularization and to initialize the RL policy, we train the architecture described in Appendix E on a dataset of roughly 46000 online Diplomacy games provided by webdiplomacy.net. We train jointly on both games with full-press Diplomacy (i.e. where players were able to communicate via messages) and no-press Diplomacy and at inference time and/or during RL, condition the relevant global feature in Table 5 to indicate the model should predict for no-press Diplomacy. Also in common with the same prior work, we apply data filtering to skip training on actions where players missed the time limit and a default null action was inputted by the website, and to only train on actions played by the top half of rated players. We also adopt the method of Jacob et al. (2022) to augment the data by permuting the labels of the 7 players randomly during training, since the game's rules are fully equivariant to such permutations. See Table 6 for a list of other hyperparameters.

## G  SELF-PLAY TRAINING

Our self-play training algorithm closely matches that of DORA from Bakhtin et al. (2021), described in detail in Section 2.3. The overall self-play procedure (see Figure 5), the training data recorded, loss function used on that data, and sampling methods we use are all the same. The differences are:

| Feature | Type | Channels |
|---|---|---|
| Season (spring/fall/winter) | One-hot | 3 |
| Year (encoded as $(y - 1901)/10$) | Float | 1 |
| Game has dialogue? | Binary | 1 |
| Scoring system used | One-hot | 2 |

Table 5: Global board state input features

| | |
|---|---|
| Learning rate | $2 \times 10^{-3}$ |
| Learning rate decay per epoch | 0.99 |
| Linear LR warmup epochs | 10 |
| Total epochs | 390 |
| Gradient clip max norm | 0.5 |
| Batch size | 16000 |
| Batches per epoch | 270 |
| Value loss weight | 0.7 |
| Policy loss weight | 0.3 |
| Optimizer | ADAM |
| Transformer encoder dropout | 0.3 |
| Policy head LSTM dropout | 0.3 |

Table 6: Hyper-parameter values used to train the IL anchor policy on human data.

- Although our model architecture is largely identical to that of past work, some minor details, including the precise encoding of input features, and the construction of the value head are different, see Appendix E for description of our architecture.

- During RL training, we initialize the RL policy proposal and value functions from the human IL anchor policy and value function (Appendix F) instead of randomly from scratch, and during training, rather than using regret matching to compute the 1-step equilibrium $\sigma$ on each turn of the game, we use DiL-piKL. The distribution of $\lambda$ and the human IL anchor policy remain fixed through all of training.

- During training, the action chosen to explore in the self-play game uses a randomly chosen $\lambda$ from the DiL-piKL distribution. Similarly, the RL policy is trained to predict the policy of a random $\lambda$. This ensures that the RL policy, when used at test time to propose actions, samples both human IL-like actions from high $\lambda$, as well as more optimized actions from lower $\lambda$, and that gamestates resulting from the entire range of possible $\lambda$ are in-distribution for the RL policy and value models.

- Unlike DORA, double-oracle action exploration is *not* used during training. We found that with the additional diversity and regularization of the human anchor policy, it was unnecessary.

- All models were also trained with the same stochastic game-end rules we used in evaluation games against human players described in Section 5.1.

- Some hyperparameters we use may be different than that of past work. See Appendix G.1 for a list of hyperparameters.

## G.1 HYPER-PARAMETERS FOR RL TRAINING

For the evaluation in this paper we trained Diplodocus and BRBot agents and re-trained DNVI, DNVI-NPU, and DORA agents. We provide the hyper-parameters used in table 7. We show parameters out of 3 agents from Bakhtin et al. (2021) as they are the same.

## G.2 INFERENCE TIME EFFECT OF DiL-piKL

In Figure 6 we show that running DiL-piKL at evaluation time alone is not enough to get the demonstrated performance improvement in population scores. Using DiL-piKL on top of human imitation-

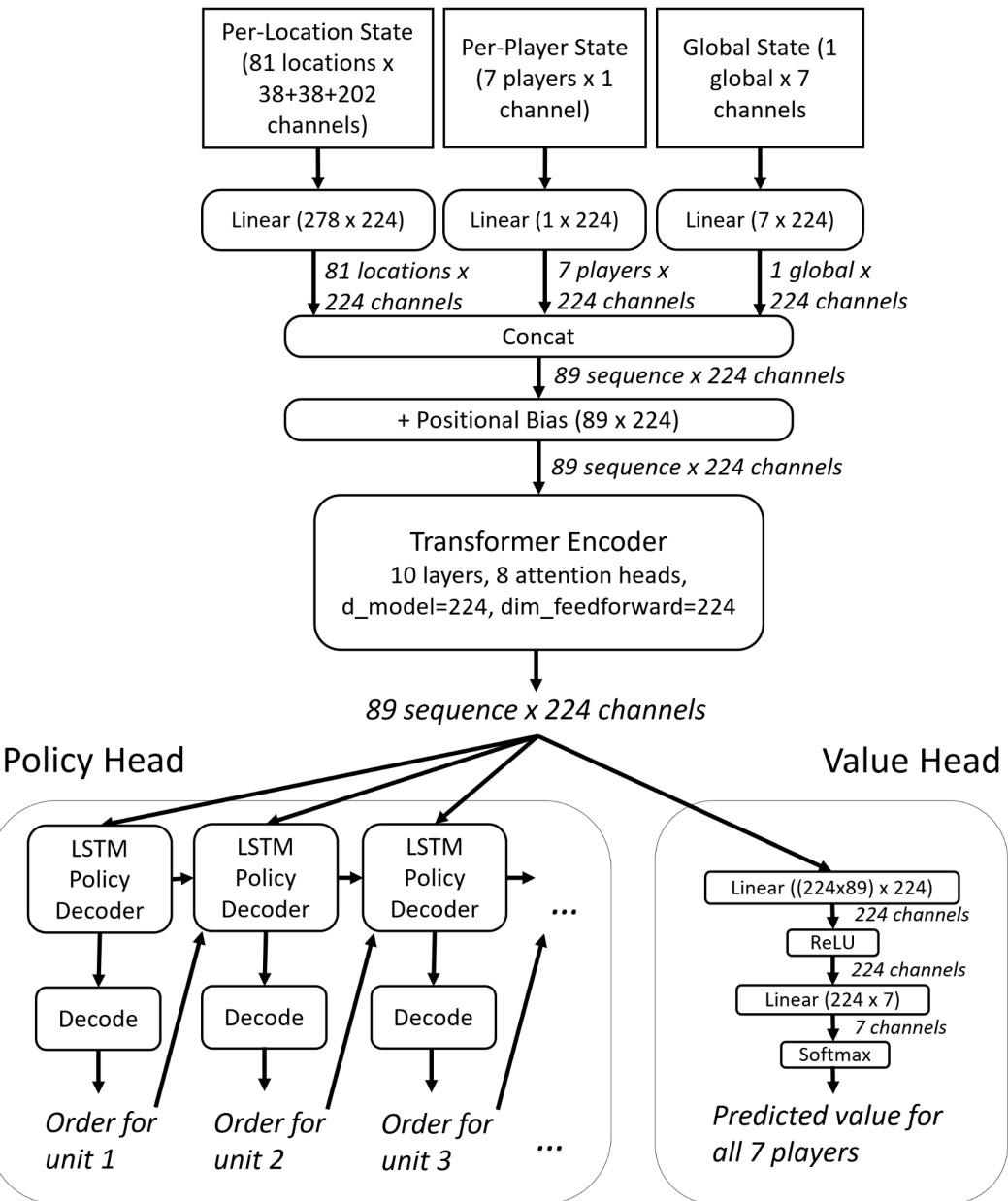

Figure 4: Model architecture used for policy/value learning in no-press Diplomacy.

learned policy/value functions does not improve the population score compared to Hedge. However, applying this search method on top of policies/values that were trained via RL with DiL-piKL results in significant improvement in the scores.

## H   BAYES-ELO

BayesElo (Coulom, 2005) models each player's expected share of the total score in a 2-player game as proportional to:

$$\exp((r_i + b_{s(i)})/c)$$

---

**Algorithm 2:** RL Loop

---

1 **function** DATAGENERATIONLOOP()
2    **while** *true* **do**
3       Game ← NEWGAME()
4       $\theta_v$ ← GETNEWVALUEFUNCTION()
5       $\theta_\pi$ ← GETNEWPOLICYFUNCTION()                         // Not used for NPU algorithm
6       **while** *not* ISDONE*(Game)* **do**
7          $s$ ← ENCODESTATE(Game)
8          $\boldsymbol{A}$ ← GETPLAUSIBLEACTION($\theta_\pi$)
9          $\boldsymbol{A}$ ← DOUBLEORACLEACTIONEXPLORATION($\boldsymbol{A}, \theta_\pi, \theta_v$)       // Only used for DORA
10         $\boldsymbol{\sigma}, \boldsymbol{u}$ ← RUNSEARCH($s, \boldsymbol{A}, \theta_v, \tau$)          // Regret Matching or DiL-piKL
11         SENDTOBUFFER($s, \boldsymbol{\sigma}, \boldsymbol{u}$)
          // Sample from the policy with possible $\epsilon$-exploration
12         $\boldsymbol{a}$ ← SELECTACTION($\boldsymbol{\sigma}$)
13         Game ← NEXTSTATE(Game, $\boldsymbol{a}$)

14 **function** TRAININGLOOP()
15    $\theta_v$ ← $BCValue$()                                       // Not used for DORA
16    $\theta_\pi$ ← $BCPolicy$()                                // Not used for DORA
17    **while** *true* **do**
18       $s, \boldsymbol{\sigma}, \boldsymbol{u}$ ← READFROMBUFFER()
19       $Loss$ ← POLICYLOSS($s, \boldsymbol{\sigma}, \theta_\pi$) + VALUELOSS($s, \boldsymbol{u}, \theta_v$)
20       GRADIENTSTEP($Loss$)
21       SAVENEWVALUEFUNCTION($\theta_v$)
22       SAVENEWPOLICYFUNCTION($\theta_\pi$)

---

Figure 5: High-level description of DNVI-style algorithms. The DORA agent is initialized from scratch and requires a Double Oracle action exploration procedure to perform well. The NPU (no policy update) modification uses the behavioral cloning policy for the policy proposal network throughout the whole training. DORA, DNVI, and DNVI-NPU use RM as the search algorithm, while the other training methods use DiL-piKL. .

| | Diplodocus High | Diplodocus Low | BRBot | DORA |
|---|---|---|---|---|
| Learning rate | | $10^{-4}$ | | |
| Gradient clip max norm | | 0.5 | | |
| Warmup updates | | 10k | | |
| Batch size | | 1024 | | |
| Buffer size | | 1,280,000 | | |
| Max train/generation ratio | | 6 | | |
| Optimizer | | ADAM | | |
| Transformer encoder dropout | | 0 | | |
| Policy head LSTM dropout | | 0 | | |
| Search algorithm | DiL-piKL | DiL-piKL | DiL-piKL | RM |
| Type distribution (self) | $\{10^{-2}, 10^{-1}\}$ | $\{10^{-4}, 10^{-1}\}$ | $\{+\infty\}$ | $\{0\}$ |
| Type distribution (other) | $\{10^{-2}, 10^{-1}\}$ | $\{10^{-4}, 10^{-1}\}$ | $\{0\}$ | $\{0\}$ |
| Search iterations | 256 | 256 | 256 | 256 |
| Number of candidate actions ($N_c$) | 50 | 50 | 50 | 50 |
| Max candidate actions per unit | 6 | 6 | 6 | 6 |
| Nash explore ($\varepsilon$) | 0.1 | 0.1 | 0.1 | 0.1 |
| Nash explore, S1901M | 0.1 | 0.1 | 0.1 | 0.3 |
| Nash explore, F1901M | 0.1 | 0.1 | 0.1 | 0.2 |

Table 7: Hyper-parameter values used to train RL agents.

where $r_i$ is the Elo rating of player $i$, $b_1$ and $b_2$ are the advantage/disadvantage of playing first/second in Elo, $s(i) \in 1, 2$ indicates whether $i$ played first or second, and $c = 400 \log_{10}(e)$ is a fixed scaling constant that adjusts for the particular arbitrary numerical scale of ratings expected by users, in

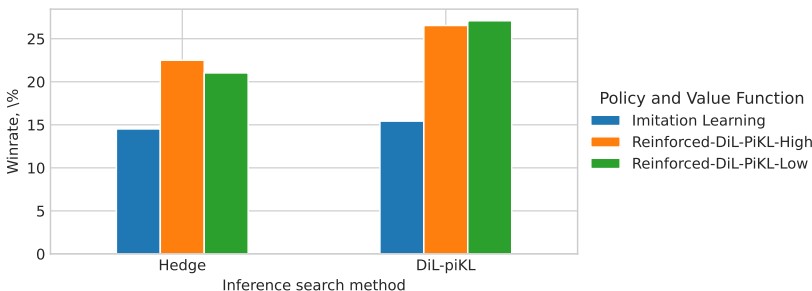

Figure 6: Performance of different search algorithms at inference time for models trained with RL and IL. Applying DiL-piKL at inference time rather than Hedge only slightly improves an IL-based search agent, but greatly improves RL agents trained with DiL-piKL.

particular that 400 points in Elo systems generally corresponds to a 10-fold increase in expected winning odds or expected average score..

It then finds joint maximum-a-posteriori values $r_i$, $b_i$ given all observed data and an optional prior to regularize the model. In some cases, the biases $b_i$ may also be hardcoded or provided as parameters rather than inferred from the data, in our work we infer them. In our application, we use a weak Bayesian prior such that each player's rating was a-priori normally distributed around 0 with a standard deviation of around 350 Elo.

BayesElo generalizes naturally to more than 2 players simply by allowing $i$ and $s(i)$ to range over $\{1, ..., n\}$ rather than $\{1, 2\}$, and we straightforwardly apply this to Diplomacy. Since there are 7 players, we similarly jointly fit $b_1, ... b_7$ on the data to model the asymmetric advantage/disadvantage of the 7 different starting positions. Computed Elo ratings closely reflect empirical winning percentages of players in a given population, but also take into account variability in the strength of opposition in a game, and the starting advantage/disadvantage $b_i$. For example, if a player achieved a high average score but was abnormally lucky in drawing advantageous starting countries across all their games, then the model would likely estimate a lower rating than if they achieved the same results with more difficult starting countries.

In Diplomacy, on the 200 games of the human tournament in which we evaluated Diplodocus and other agents, the empirical fitted $b_i$ values for the 7 different starting countries in the game are displayed in Table 8.

| Starting Country | $b_i$ (Elo) |
|---|---|
| Austria | -24 |
| England | -43 |
| France | 59 |
| Germany | 18 |
| Italy | -21 |
| Russia | -16 |
| Turkey | 27 |

Table 8: For each starting country, the empirical advantage/disadvantage of starting as that country measured in Elo rating equivalent units, fitted jointly with all players' Elo ratings on the 200 Diplomacy games of the tournament. The values roughly agree with common opinions among Diplomacy players, particularly that France is one of the best starting countries in no-press, while Austria and England are among the weaker starts.

