# OpenReview forum: "Mastering the Game of No-Press Diplomacy via Human-Regularized Reinforcement Learning and Planning"
_ICLR.cc/2023/Conference — ICLR 2023 notable top 5%_

### Official Review · Reviewer_qUpu · 2022-10-21

**Confidence:** 4
**Correctness:** 3
**Technical Novelty And Significance:** 3
**Empirical Novelty And Significance:** 3
**Recommendation:** 8

**Clarity, Quality, Novelty And Reproducibility:**

This paper is exceptionally clear and well written. It contains a novel result with theoretical and empirical (quantitative & qualitative) results.

This work is likely impossible to reproduce without the code being released; but, moreover, it contains human evaluation.

**Strength And Weaknesses:**

**Strengths**
 - This paper is exceptionally well written. It provides an accurate and concise narrative of all modern work (ie., those containing DRL; it would be nice if they referenced some pre-DRL work in this space) in a very easy-to-process exposition; especially, as it pertains to the motivation behind each innovation.
 - The innovation focuses on a real problem: fixed-coefficients in compound losses limiting the scope of discoverable strategies. The propose a reasonable solution of sampling throughout training and theoretically show that in certain cases this maintains convergence garuntees. The convergence garuntees are for their modified objective and not the true objective, and I would be interested to hear the author's thoughts on why this is reasonable and how their work might also show convergence to the true objective? The true objective should be to learn an equilibrium (maximize utility), and the human-like component of the loss should only seek to influence selection from equilibrium.
 - The paper also contains empirical results on full no-press Diplomacy with agents and humans including qualitative results. They perform notably strong against within the agent population, but the quantitative results including humans are inconclusive. The qualitative results are mixed, but generally positive towards the proposed agents.  While the results are not overwhelmingly positive it's good to see such diverse and interesting analysis. I would suggest that the authors also consider in future work inquiring from human-players about finesse or attempts to identify the agent player during play. Perhaps have some probability that no agents are present?

**Weaknesses**
 - A large concern I've had with this work and its predecessors is that it concerns the equilibrium selection problem without adequate discussion of its tradeoffs. This work penalizes an agent towards behaviors similar to a single representative policy that are meant to represent an entire population of interest. Artifacts of construction of the representative also play a large role in the equilibrium selection process by are not discussed. Why should a single policy be sufficient to represent a target population of interest? What about biases about the representation of individuals within the population as it pertains to the representative? How would this effect performance across diverse sub-sets of the target population and to those outside of the population?
 - Could the authors please elaborate on why the beta distribution should be considered common knowledge? (Sec 3, Par 2).
 - This work proposes effectively using a distribution over behaviour policies to generate training data. I am curious how the authors arrived at the coefficients that represent the types within this population? Moreover, how diverse are these policies? I should expect that adding diverse experiences would cause the agent to require much longer training, but Fig3 makes it seem that the agents require effectively no training?
 - As the authors are aware the country played has a large impact on the success of an agent (Appendix). This makes comparing agents on such a small number of games challenging when attempting to normalize out the power that they played. The Elo bias numbers associated with each power that the authors construct are so large that would render any attempt to understand Table 2 unfair. I would have preferred that more controlled studies were performed to really try and understand the differences with the least confounding variables. I do appreciate that human studies are very expensive and the results within are still nice.
 - The claim of "Mastery" is exceptionally strong considering the inconclusive evidence about this method being measurably different from vastly simpler baselines.
 - The main methodological change is related to influencing the human-likeness coefficient, but the experimental results do not focus on trying to understand the changes, but instead solely care about a particular game.
 - It's not clear to mean that being human-predictable is a good objective to optimize towards. Consider what Expert 1 said (pg13): "I think it is quite predictable in what motivates ... That's potentially exploitable, even with the strong tactics". This is a very astute observation, and I tend to agree with the underlying idea. We would like to select human-like equilibria for cooperative, but this may hinder the policy's performance when cooperation is no longer needed. A particularly interesting problem in Diplomacy where the amount of cooperation required is fluid over the course of the game.

**Summary Of The Paper:**

Left to their own devices agents will learn equilibrium specific to the coplayer(s) that are encountered throughout their training algorithm. This can be problematic in (semi-)cooperative games where successful equilibrium selection depends on a population of players distinct from those that were featured during training. This manuscript investigates this problem by attempting to learn "human-like" equilibrium selection in the many-player general-sum game of no-press Diplomacy. They extend a previous line of work that proposed an optimization objective of a weighted combination of being performant and human-like. They consider generalizing the fixed weighting into a sampling process so that experience is generated on a spectrum of performant and human-like. They test a suite of agents on a set of paid self-volunteered humans of indeterminant strength (likely quite good in certain variants of Diplomacy; note each variant is very different). Their proposed agent comparable similar to the best human in the population alongside a simple BR(BC) baseline.



**Summary Of The Review:**

This paper presents an improvement for the human-like equilibrium selection problem. It demonstrates its efficiency with a in-depth study of the game of no-press Diploamcy. The results are not convincing enough that mastery was achieved, but the investigation is sound and this is a noteworthy milestone towards solving Diplomacy.

Therefore I think it is worthy of publication, but I strongly encourage the authors to tone down the claims of its accomplishments, because it undermines the work and credibility.

---

> ### Author Response · Authors · 2022-11-14
> **Response to Reviewer qUpu**
>
> Thank you for your helpful feedback!
>
> >A large concern I've had with this work and its predecessors is that it concerns the equilibrium selection problem without adequate discussion of its tradeoffs. This work penalizes an agent towards behaviors similar to a single representative policy that are meant to represent an entire population of interest. Artifacts of construction of the representative also play a large role in the equilibrium selection process by are not discussed. Why should a single policy be sufficient to represent a target population of interest? What about biases about the representation of individuals within the population as it pertains to the representative? How would this effect performance across diverse subsets of the target population and to those outside of the population?
>
> Note that here the policy here is *not* merely a representative player policy. Rather it is a prediction of the overall average probability distribution of actions expected, where the average is across the human players observed in the training data conditional on the game state/history (including any information that within-episode history may reveal about those players). This is a natural consequence of the fact that the player identities are not labeled during training. So this is a policy in the sense that it is a mapping from (state/history, action) -> probability of that action, but it is not the policy of any individual agent, it is a prediction of the population as a whole. Since we assume players are anonymous during play, the conditional average population action distribution is the same as the empirical action distribution one anticipates from players given the Bayesian uncertainty about who those players are.
>
> Regarding biases - yes, the human players in the training data may not precisely match the actual test-time player distribution. Diplodocus would likely perform worse in a hypothetical isolated population of players who among themselves developed conventions different than the broader Diplomacy community but had otherwise a similar distribution of skills. In our case, our data was from webDiplomacy.net which is a popular online site in the broad community and the tournament for testing consisted similarly of a wide range of players.
>
> >Could the authors please elaborate on why the beta distribution should be considered common knowledge? (Sec 3, Par 2).
>
> The purpose of making it common knowledge is to model the fact that players do not know the agent’s type. For example, at the start of the game they do not know that we are a strong agent (because all players are anonymous), we know they don’t know that we are a strong agent, etc. If, for example, it were common knowledge at the start of the game that the agent was playing according to a specific lambda policy, then the common-knowledge beta distribution for the agent could be set to 100% probability on that lambda value.
>
> >This work proposes effectively using a distribution over behavior policies to generate training data. I am curious how the authors arrived at the coefficients that represent the types within this population? Moreover, how diverse are these policies?
>
> The lambda distribution coefficients were hand-selected based on human intuition about the game. We think that finding more principled ways of selecting the lambda distribution would be an interesting and useful direction for future work. We will add this point to the discussion section. Our goal was to primarily capture diversity through the human data rather than through the distribution of lambda parameters. The use of DiL-piKL results in a relatively fatter-tailed distribution over actions compared to piKL.
>
> >I should expect that adding diverse experiences would cause the agent to require much longer training, but Fig3 makes it seem that the agents require effectively no training?
>
> Diplodocus and DNVI do require training, but train much faster than DORA because they are regularized towards a human policy, whereas DORA needs an enormous number of updates to learn reasonable strategies in a combinatorially large state and action space from scratch. The figure is a bit misleading; we will adjust it to show score at earlier time points.

---

> > ### Author Response · Authors · 2022-11-14
> > **Response to Reviewer qUpu (continued)**
> >
> > >The claim of "Mastery" is exceptionally strong considering the inconclusive evidence about this method being measurably different from vastly simpler baselines.
> >
> > We considered a few different options for the title and felt that “Mastering” was a weaker claim than “Expert-Level Performance” due to the precedent set in the original AlphaGo paper (“Mastering the Game of Go with Deep Neural Networks and Tree Search”) of beating the European Go champion even though there were far stronger humans that it was not shown to be able to beat. While the human experiments are high-variance, we felt that a combination of the human experiments, the human subjective evaluation, and the agent population results consistently pointed toward Diplodocus being among the strongest players for this format of the game. We do not intend to imply that this paper should be the final word on research into no-press Diplomacy, any more than the original AlphaGo paper was the final word on Go.
> >
> > >It's not clear to mean that being human-predictable is a good objective to optimize towards. Consider what Expert 1 said (pg13): "I think it is quite predictable in what motivates ... That's potentially exploitable, even with the strong tactics". This is a very astute observation, and I tend to agree with the underlying idea. We would like to select human-like equilibria for cooperative, but this may hinder the policy's performance when cooperation is no longer needed. A particularly interesting problem in Diplomacy where the amount of cooperation required is fluid over the course of the game.
> >
> > While we do assume that all players play close to human policy to learn the value function during self-play RL training, we do not make such an assumption for our agent during actual play. Instead, our agent plays a policy that corresponds to a very low lambda, i.e., a policy that maximizes expected value given the predicted policies of others.
> >
> > >This work is likely impossible to reproduce without the code being released; but, moreover, it contains human evaluation.
> >
> > Thanks for raising this issue. We are preparing all code and models for release, and they will be available by the end of November.

---

> > > ### Comment · Reviewer_qUpu · 2022-11-17
> > > **Response**
> > >
> > > Thank you for answering all of my questions. I enjoyed reading the work and I hope it continues more into the future.
> > >
> > > While this work is entirely fit for publication and I would not attempt to stand in its way for publication, I would like to re-advocate two points:
> > >   * Inclusion of commentary about equilibrium selection; specifically, that this selects human-like equilibria and there are pros and cons to this choice. Perhaps this may simply just be contained within the ethics section.
> > >   * Weaker claims than "Mastery" or similarly braggadocious terms. The authors are correct to point-out that previous papers centered around beating expert-level human players have used this term; however, I don't think that's a great reason to justify the same choice. Mastery historically implies that there are a set of goals/tasks and the agent needs to show proficiency in all of them. Without specifying the set of tasks, for a game, one expects that the task is all coplayers over all game settings. Effectively solving the game. "Expert-Level Performance" feels more apt to me in accurately framing the problem tackled within this work: as you are literally modelling off expert performance and make no claim about any other player category. I am not saying that this is the best title, nor that I have a great alternative. I apologize for going on about this, but big claims can make the publication of future work subject to additionally challenging, and there's still so much we can learn from No-Press Diplomacy. So where possible, I hope the author's can ease language to pave an easier way for others to build upon their work and explore diverse avenues.
> > >
> > > Thank you for entertaining my proposition.

---

### Official Review · Reviewer_gn6m · 2022-10-24

**Confidence:** 3
**Correctness:** 4
**Technical Novelty And Significance:** 3
**Empirical Novelty And Significance:** 3
**Recommendation:** 8

**Clarity, Quality, Novelty And Reproducibility:**

The paper is well written. The approach is mostly combination of existing techniques, but the strong performance of the proposed algorithm make the paper a significant contribution. The results are probably difficult to reproduce, which is not unexpected due to the complexity of the algorithm.



**Strength And Weaknesses:**

The proposed algorithm is sound. It has some novelty, although it is mostly a combination of existing algorithms. There is also a theoretical analysis of the DiL-PiKL component.

The empirical performance seems also impressive, and the expert analysis certainly sheds further insights into the strength and weaknesses of the algorithm variants.

It is clear that regularization is mainly added to inject some cooperation flavor into the algorithm. A large chunk of the human play does not involve cooperating moves, but moves that aim to improve the position of the player (maximize the reward). Because regularization is uniform across all moves, it is forced to be small enough to avoid drop in performance (even if the human moves are reasonably good), which leads to weaker and less coherent cooperation, when needed (as indicated by the experts).


**Summary Of The Paper:**

The paper considers multi-player games, in particular No-press diplomacy. A new algorithm is proposed that combines self-play reinforcement learning with a no-regret learning algorithm that is regularized by human play.

Theoretical properties of the no-regret algorithm are proven bounding the cumulative regret for the multi-player case, and showing convergence to Bayes-Nash equilibrium in the two-player case.

Experiments with human players show that variants of the algorithm perform better than some baseline algorithms, and even on par with the expert human players included in the tournament. The human analysis of the computer players' behavior reveals that the proposed variants elicit more human-like cooperation strategies.


**Summary Of The Review:**

The paper proposes an algorithm with theoretical guarantees and strong empirical performance in a difficult and complex domain. While the components of the algorithm are not very novel, the paper still provides a significant contribution.

---

> ### Author Response · Authors · 2022-11-14
> **Response to Reviewer gn6m**
>
> Thank you for your helpful feedback! We appreciate your positive comments about the paper.

---

### Official Review · Reviewer_FjZ3 · 2022-10-26

**Confidence:** 3
**Correctness:** 4
**Technical Novelty And Significance:** 3
**Empirical Novelty And Significance:** 3
**Recommendation:** 8

**Clarity, Quality, Novelty And Reproducibility:**

Clarity
  * The paper is well written. The ideas are well structured and nicely detailed.

Quality
  * The paper is technically sound. It builds on prior work. The proposed ideas are shown to be effective in experiments. I think this paper would be a nice addition to the literature on agent design in games requiring cooperation with humans.

Novelty
  * The paper contains a number of technical contributions. Diplodocus is a novel algorithm, as far as I can tell.

Reproducibility
  * There seems to be sufficient detail included. Including code would have been better.

**Strength And Weaknesses:**

Strengths
  * The paper tackles a problem that is less well-studied (compared to typical single-agent environments.)
  * The paper builds on previous works methodically. The proposed algorithm is intuitively clear. Each idea is well-motivated and the overall algorithm is easy to follow.
  * The experiments are well-constructed and nicely detailed. A number of strong baselines are used.
  * The overall algorithm, Diplodocus, clearly improves the state-of-the-art.

Weaknesses
  - I didn't spot any major technical issues with the paper. A couple of minor suggestions and questions.
  - How might a standard MCTS (AlphaGo style) fare? While unsound for the reasons mentioned, it might make the experimental section more complete.
  - What would be needed to make the error bars in Table 2 smaller?
  - I'd be curious to better understand how performance varies as a function of human player level. For example, can Diplodocus outperform weak human players? This would likely require a more complex tournament setup and is likely outside the scope of the paper.


**Summary Of The Paper:**

The paper proposes new algorithms (DiL-piKL, RL DiL-piKL, Diplodocus) for decision-making in games requiring cooperation with humans. The proposed method combines a number of ideas (self-play, leveraging imitation learning with KL-divergence regularization) to produce an agent that learns to play well against (and in-cooperation with) human players in a tournament of no-press diplomacy. Theoretical justification (bounded regret under certain assumptions) is included. The experimental results show the agent performs well compared to human players, measured by final rank as well as subsequent analysis of gameplay.

**Summary Of The Review:**

The paper contains a new algorithm for training agents that can play well in games requiring cooperation with other human players. It seems to extend the state-of-the-art in this area although the error bars are a bit too large. The paper is well written and is technically sound.

---

> ### Author Response · Authors · 2022-11-14
> **Response to Reviewer FjZ3**
>
> Thank you for your helpful feedback!
>
> >How might a standard MCTS (AlphaGo style) fare? While unsound for the reasons mentioned, it might make the experimental section more complete.
>
> MCTS has traditionally not been a baseline for imperfect-information and/or simultaneous-move games due to its lack of soundness in these settings.
>
> >What would be needed to make the error bars in Table 2 smaller?
>
> The error bars in Table 2 are the standard error of the mean score across tournament games with human participants. Reducing the error bars would require playing more games with human participants. Since each game took 6 humans roughly 3 hours, it would be difficult to collect more games per player, especially for the human participants.
>
> >I'd be curious to better understand how performance varies as a function of human player level. For example, can Diplodocus outperform weak human players? This would likely require a more complex tournament setup and is likely outside the scope of the paper.
>
> Typically in Diplomacy tournaments and in online website rankings, rank is measured based on a player’s performance in games with other players of diverse skill levels. This is intentional, because a big part of Diplomacy is effectively modeling and either cooperating with or exploiting suboptimal players. Therefore, we felt the most appropriate way to evaluate the bot was to compare Diplodocus’s performance in this diverse population to that of expert humans. Additionally, measuring performance against specific subpopulations of the player pool in a meaningful way would indeed require many more human games and be prohibitively expensive.

---

### Comment · Program_Chairs · 2023-03-21
**Congratulations on winning the ICLR 2023 Outstanding Paper Honorable Mention!!**

Hello authors,

Big congratulations! For more details about the award process this year, please see https://blog.iclr.cc/2023/03/21/announcing-the-iclr-2023-outstanding-paper-award-recipients/

Best,

Mengdi, on behalf of ICLR 2023 PCs

---

### Decision · Program_Chairs · 2023-01-20

**Decision:**

Accept: notable-top-5%

**Justification For Why Not Higher Score:**

N/A

**Justification For Why Not Lower Score:**

There is no major issue. The performance on no-press diplomacy games is inspiring. That being said, spotlight is also OK.

**Metareview: Summary, Strengths And Weaknesses:**

This paper proposes a new algorithm for multi-player games. The algorithm combines self-play reinforcement learning with a no-regret learning algorithm that is regularized by human play. The new algorithm achieves performance that is comparable or even better than the best human in no-press diplomacy games. A theoretical guarantee in the simple setting is provided, and a solid empirical results is given. All reviewers strongly recommend the acceptance of this paper.

**Note From Pc:**

if the above contains the word "oral" or "spotlight" please see: "oral" presentation means -> notable-top-5% and "spotlight" means -> notable-top-25%. As stated in our emails, we are disassociating presentation type from AC recommendations